



# Atmospheric photooxidation and ozonolysis of $\Delta^3$-carene and 3-caronaldehyde: Rate constants and product yields

Luisa Hantschke[1], Anna Novelli[1], Birger Bohn[1], Changmin Cho[1], David Reimer[1], Franz Rohrer[1], Ralf Tillmann[1], Marvin Glowania[1], Andreas Hofzumahaus[1], Astrid Kiendler-Scharr[1], Andreas Wahner[1], and Hendrik Fuchs[1]

[1]Institute of Energy and Climate Research, IEK-8: Troposphere, Forschungszentrum Jülich GmbH, Jülich, Germany

**Correspondence:** Hendrik Fuchs
(h.fuchs@fz-juelich.de)

**Abstract.** The oxidation of $\Delta^3$-carene and one of its main oxidation products, caronaldehyde, by the OH radical and $O_3$ was investigated in the atmospheric simulation chamber SAPHIR under atmospheric conditions for $NO_x$ mixing ratios below $2\,\mathrm{ppbv}$. Within this study, the rate constants of the reaction of $\Delta^3$-carene with OH and $O_3$, and of the reaction of caronaldehyde with OH were determined to be $(8.0 \pm 0.5) \times 10^{-11}\,\mathrm{cm^3 s^{-1}}$ at $304\,\mathrm{K}$, $(4.4 \pm 0.2) \times 10^{-17}\,\mathrm{cm^3 s^{-1}}$ at $300\,\mathrm{K}$ and $(4.6 \pm 1.6) \times$

5   $10^{-11}\,\mathrm{cm^3 s^{-1}}$ at $300\,\mathrm{K}$, respectively, in agreement with previously published values. The yields of caronaldehyde from the reaction of OH and ozone with $\Delta^3$-carene were determined to be $(0.30 \pm 0.05)$ and $(0.06 \pm 0.02)$, respectively. Both values are in reasonably well agreement with reported literature values. An organic nitrate ($RONO_2$) yield from the reaction of NO with $RO_2$ derived from $\Delta^3$-carene of $(0.25 \pm 0.04)$ was determined from the analysis of the reactive nitrogen species (NOy) in the SAPHIR chamber. The $RONO_2$ yield of the reaction of NO with $RO_2$ derived from the reaction of caronaldehyde with OH

10   was found to be $(0.10 \pm 0.02)$. The organic nitrate yields of $\Delta^3$-carene and caronaldehyde oxidation with OH are reported here for the first time in the gas phase. An OH yield of $(0.65 \pm 0.10)$ was determined from the ozonolysis of $\Delta^3$-carene. Calculations of production and destruction rates of the sum of hydroxyl and peroxy radicals ($ROx = OH + HO_2 + RO_2$) demonstrated that there were no unaccounted production or loss processes of radicals in the oxidation of $\Delta^3$-carene for conditions of the the chamber experiments. In an OH free experiment with added OH scavenger, the photolysis frequency of caronaldehyde was

15   obtained from its photolytical decay. The experimental photolysis frequency was a factor of 7 higher than the value calculated from the measured solar acintic flux density, an absorption cross section from the literature and an assumed effective quantum yield of unity for photodissociation.





## 1 Introduction

On a global scale, the emission of carbon from biogenic volatile organic compounds (BVOC) exceeds 1000 Tg per year (Guen-
ther et al., 2012). Among all BVOC emissions, monoterpenes are the second most important class of species, contributing up
to $16\,\%$ to the total emissions. As they are unsaturated and highly reactive, knowledge of their atmospheric chemistry is crucial
to understand the formation of secondary pollutants such as ozone ($O_3$) and particles (e.g Seinfeld and Pandis, 2006; Zhang
et al., 2018).

Of the total global annual monoterpene emissions, $\Delta^3$-carene contributes $4.5\,\%$ making it the $7^{\text{th}}$ most abundant monoter-
pene species (Geron et al., 2000). $\Delta^3$-carene is primarily emitted by pine trees. Measured emission rates are up to $(85\,\pm\,$
$17)\,\mathrm{ngg(dw)}^{-1}\mathrm{h}^{-1}$ (nanograms monoterpenes per gram dry weight (dw) of needles and hour) from Scots pines (Komenda
and Koppmann, 2002) and $57\,\mathrm{ngg(dw)}^{-1}\mathrm{h}^{-1}$ from maritime pine. Therefore, $\Delta^3$-carene regionally gains in importance, for
example in boreal forests and the mediterranean region. Hakola et al. (2012) measured the mixing ratios of monoterpenes over
a boreal forest in Hyytiälä, Finland, and found $\Delta^3$-carene to be the second most abundant monoterpene after $\alpha$-pinene. How-
ever, while the atmospheric chemistry of some monoterpenes such as $\alpha$-pinene or $\beta$-pinene has been investigated in a number
of experimental and theoretical studies (e.g. Peeters et al. (2001); Aschmann et al. (2002b); Eddingsaas et al. (2012); Rolletter
et al. (2019) and references therein), only few studies exist investigating the photooxidation of $\Delta^3$-carene.

The similarity of its structure and type of products, e.g. caronaldehyde and pinonaldehyde that have been reported in literature
(e.g. Arey et al. (1990); Hakola et al. (1994); Yu et al. (1999)) suggest that $\Delta^3$-carene behaves similar like $\alpha$-pinene (Colville
and Griffin, 2004) regarding its oxidation mechanism. The structural relations between $\Delta^3$-carene and $\alpha$-pinene and their oxi-
dation products caronaldehyde and pinonaldehyde are shown in Fig. 1.

In the atmosphere, monoterpenes mainly react with the hydroxyl radical (OH) and ozone ($O_3$) during daytime and $O_3$ and the
nitrate radical ($NO_3$) during nighttime. From the reaction of $\Delta^3$-carene with OH, organic peroxy radicals ($RO_2$) are formed
that can subsequently react with NO regenerating either eventually OH or forming organic nitrates ($RONO_2$). Organic nitrates
terminate the radical chain. They are typically less volatile and more hydrophilic than their parent organic compound making
them important for secondary organic aerosol (SOA) formation. Recent measurements suggest that 17 to $23\,\%$ of organic aer-
sols contain molecules with nitrate functional groups (Rollins et al., 2013). $RONO_2$ species serve also as NOx (NOx = $NO_2$
+NO) reservoirs, hereby influencing e.g. $O_3$ formation (Farmer et al., 2011).

In oxidation product studies, caronaldehyde was found to be the main daytime organic oxidation product from the oxidation
of $\Delta^3$-carene (Arey et al., 1990; Reissell et al., 1999). The vapor pressure of this compound is lower than the vapor pressure
of $\Delta^3$-carene, possibly making it a contributor to particle formation in the atmosphere (Yokouchi and Ambe, 1985; Hallquist
et al., 1997).

In the present study, the following reactions of $\Delta^3$-carene were investigated in a series of experiments in the atmospheric sim-
ulation chamber SAPHIR:





**Figure 1.** Structures of $\Delta^3$-carene and its oxidation product caronaldehyde, and $\alpha$-pinene and its oxidation product pinonaldehyde.


$$\Delta^3-\text{carene} + \text{OH} \quad \rightarrow \quad \text{prod.} \tag{R1}$$

$$\Delta^3-\text{carene} + \text{O}_3 \quad \rightarrow \quad \text{prod.} + \text{OH} \tag{R2}$$

$$\text{caronaldehyde} + \text{OH} \quad \rightarrow \quad \text{prod.} \tag{R3}$$

$$\text{caronaldehyde} + h\nu \quad \rightarrow \quad \text{prod.} \tag{R4}$$

The reaction rate constants of Reactions R1 - R3 and the photolysis frequency of Reaction R4 are determined from the experimental decay of the respective VOC and compared to previously published values. Furthermore the OH yield of Reaction R2 and the caronaldehyde yield of Reactions R1 and R2 are determined under atmospheric conditions. The organic nitrate yields of the reactions of NO with RO$_2$ radicals formed in Reactions R1 and R3 are obtained for the first time in the gas phase. Radical and trace gas measurements are used to analyse, if unaccounted radical reactions are of importance in the oxidation
mechanism of $\Delta^3$-carene.

## 2 Methods

### 2.1 Experiments in the SAPHIR chamber

SAPHIR is a double-walled atmospheric simulation chamber made of Teflon (FEP) film. It is of cylindrical shape (5 m diameter, 18 m length) with an innner volume of 270 m$^3$. A shutter system allows for experiments to be conducted both in the dark



**Table 1.** Experimental conditions for all discussed oxidation experiments. Concentrations are given for the conditions in the SAPHIR chamber at the time of the first VOC injection, VOC concentrations refer to the first injection.

| Expt. No | Date | Type of experiment | RH / % | NO / ppbv | $O_3$/ ppbv | VOC / ppbv | T / K |
|---|---|---|---|---|---|---|---|
| E1 | 13 June 2019 | $\Delta^3$-carene, OH oxidation, low $NO_x$ | 60 | 0.8 | 35 | 5 | 303 |
| E2 | 21 May 2020 | $\Delta^3$-carene, OH oxidation, low $NO_x$ | 40 | 0.3 | 90 | 6 | 305 |
| E3 | 31 May 2020 | $\Delta^3$-carene, ozonolysis, dark chamber | 20 | 0.1 | 120 | 7 | 300 |
| E4 | 31 May 2020 | $\Delta^3$-carene, ozonolysis, dark chamber[a] | 20 | 0.1 | 120 | 7 | 300 |
| E5 | 27 May 2020 | caronaldehyde, photolysis[a] | 40 | $\leq 1.5$ | 80 | 17 | 300 |
| E6 | 27 May 2020 | caronaldehyde, OH oxidation | 40 | $\leq 1.5$ | 80 | 17 | 300 |

[a] $\sim 100$ ppmv CO added as OH scavenger

(shutters closed) and sunlit (shutters opened) chamber. With the shutters opened, the chamber is exposed to sunlight, therefore allowing for natural photooxidation conditions. The pressure of the chamber is kept 35 Pa higher than the outside pressure to avoid leakage from ambient air into the chamber. Ultra pure nitrogen and oxygen (Linde, purity 99.99990 %) are used to mix the synthetic air for the experiments. The space between the two FEP films is permanently flushed with ultra-pure nitrogen. In order to compensate for the air lost to small leakages and sampling from the instruments, a flow of synthetic air is continuously

introduced into the chamber leading to a dilution of trace gases of about 4 % per hour. OH radicals are produced mainly from nitrous acid (HONO) photolysis. HONO is formed on the chamber walls upon illumination of the chamber. The photolysis of HONO also leads to the formation of nitric oxide (NO), so that the $NO_x$ concentration increases continuously over the course of an experiment. A more detailed description of the SAPHIR chamber and its features can be found elsewhere (e.g. Rohrer et al. (2005)).

In total, six experiments were conducted to study the OH induced photochemical degradation and ozonolysis of $\Delta^3$-carene and the photolysis and photooxidation of caronaldehyde under different conditions. Experimental conditions for each experiment are summarized in Table 1.

Two experiments, experiment E1 and experiment E2, were conducted to investigate the OH reaction of $\Delta^3$-carene with NO

mixing ratios of less than 0.8 ppbv. Additionally, the ozonolysis of $\Delta^3$-carene was studied under dark conditions (chamber roof closed) in two experiments, one without (experiment E3) and one with (experiment E4) the addition of an OH scavenger. Two experiments focused on the photolysis (experiment E5) and reaction with OH (experiment E6) of caronaldehyde with NO mixing ratios below 2 ppbv.

Before each experiment, the chamber was flushed with synthetic air until the concentrations of trace gases were below the

detection limit of the instruments. At the beginning of the experiments, the chamber air was humidified by boiling Milli-Q water and adding the water vapor together with a high flow of synthetic air into the chamber. Relative humidities (RH) for





the photooxidation experiments were around 80 % at the beginning of the experiment (water vapor mixing ratio $\sim 2\,\%$), but decreased during the day mostly due to a rise in temperature inside the chamber. Values at the end of the experiment were typically 35 % relative humidity (water vapor mixing ratio $\sim 0.7\,\%$). For the ozonolysis experiment, relative humidity was

30 % at the beginning of the experiment. For some of the photooxidation experiments, ozone produced by a discharge ozonizer was injected into the chamber to reach mixing ratios of up to 60 ppbv to supress NO in its reaction with $O_3$. Up to $100\,\text{ppmv}$ of CO was injected into the chamber as an OH scavenger for the experiments E4 and E5 to study the ozonolysis of $\Delta^3$-carene and the photolysis of caronaldehyde under OH-free conditions.

During the first $2\,\text{hours}$ after opening the roof, the 'zero-air' phase, no further reactive species were added allowing for char-

acterizing chamber sources of e.g. nitrous acid (HONO), formaldehyde (HCHO) and acetone, and to quantify the background OH reactivity. The sum of OH reactant concentrations ([X]) weighted by their reaction rate coefficient with OH ($k_{OH+X}$) is called OH reactivity ($k_{OH}$). OH reactivity observed in the SAPHIR chamber in the 'zero-air' phase cannot be attributed to a specific species and is therefore seen as background reactivity. Background OH-reactivity in the range of $0.5\,\text{s}^{-1}$ is commonly observed in the SAPHIR chamber due to contaminants coming from the chamber wall upon humidification or irradiation. For

the experiments in this study, a higher background reactivity of up to $5\,\text{s}^{-1}$ was observed possibly due to a higher contamination of the chamber caused by previous experiments. Specific contaminants can remain in the chamber even after flushing the chamber with a high flow of ultra-pure synthetic air, because they are adsorbed on the chambers Teflon film. Upon humidification or illumination of the chamber, these contaminants can be desorbed resulting in a higher than usual background OH reactivity. After the injections of the VOCs, OH reactivity was dominated by their reaction with OH and the background

reactivity could be well described from measurements in the 'zero-air' phase, so that its influence is negligible for the analysis in this work, which concentrates on the times right after the injection of the VOC, when the total OH reactivity was dominated by the VOC.

Following the zero air phase, $\Delta^3$-carene (Fluka, purity 99 %) was injected two to three times as a liquid into a heated volume. The vapour was transported into the SAPHIR chamber with the above mentioned replenishment flow of synthetic air.

$\Delta^3$-carene mixing ratios ranged from 3.1 to 6.0 ppbv in the chamber right after the injections.

Caronaldehyde was synthesized by ChiroBlock GmbH (Wolfen, Germany) as an enantionmeric mixture of (+)- and (-)-caronaldehyde with a purity of >95 % determined by $^1$H-NMR measurements. The liquid was dissolved in dichlormethane (DCM) for stability reasons. The concentration of caronaldehyde in DCM was 0.5 mol/l. To exclude influences of the solvent on the chemistry observed in the experiments, additional test experiments were conducted in which only DCM was injected into the

chamber. It was found that DCM did not interfere with the measurements and that DCM chemistry did not play a role for the analysis here. Caronaldehyde was injected as a liquid dissolved in DCM and evaporated like done for liquid $\Delta^3$-carene. Caronaldehyde mixing ratios in the chamber were 5 to 9 ppbv right after the injections. Potential wall loss of caronaldehyde was determined in a test experiment and was found to be negligible on the timescale of the experiments. Therefore, wall loss of caronaldehyde was not considered in the analysis of experiments.





## 2.2 Instrumentation

$\Delta^3$-carene and caronaldehyde were detected by aproton-transfer-reaction time-of-flight mass spectrometer instruments (VOCUS-PTR-MS, Aerodyne). Both instruments were calibrated for $\Delta^3$-carene. The measured concentrations were compared to the rise of the OH reactivity measurements when $\Delta^3$-carene was injected. It was found that the $\Delta^3$-carene concentration was underestimated by the VOCUS by a factor of 3 throughout all experiments. Therefore, the concentrations were scaled, so that the measured rise of $\Delta^3$-carene by these mass spectrometric methods matched the increase in the measured OH reactivity at the time of the injections using a $\Delta^3$-carene + OH reaction rate constant of $8.0 \times 10^{-11}$ cm$^3$s$^{-1}$. The uncertainty of this calibration correction is given by the uncertainty in the $\Delta^3$-carene + OH reaction rate constant and the uncertainty in the OH reactivity measurements that is 10 % (Fuchs et al., 2017a). Pinonaldehyde was used as a substituent to calibrate the instrument for caronaldehyde, as there is no calibration standard for caronaldehyde. The concentrations obtained from this calibration were in good agreement with the rise of the OH reactivity during caronaldehyde photooxidation experiments using a reaction rate constant of $4.1 \times 10^{-11}$ cm$^3$s$^{-1}$.

Formaldehyde (HCHO) was measured by a Hantzsch monitor (AL4021, AeroLaser GmbH) and with a cavity ring-down instrument (G2307, Picarro). On average, the concentrations measured by both instruments agreed within 15 % (Glowania et al., 2021). HONO concentrations were measured by a custom-built long path absorption photometer (LOPAP) (Kleffmann et al., 2002; Li et al., 2014). NO and NO$_2$ were measured using a chemiluminescence instrument (Eco Physics) equipped with a blue-light photolytic converter for the conversion of NO$_2$ to NO. CO and water vapor was measured with a cavity ring-down instrument (G2401, Picarro) and O$_3$ with an UV absorption instrument (Ansyco). Total and diffuse spectral actinic flux densities measured by a spectral radiometer outside of the chamber were used to calculate photolysis frequencies (j) following Equation (1):

$$j = \int \sigma(\lambda)\phi(\lambda)F_\lambda(\lambda)d\lambda \tag{1}$$

with $\sigma$ the absorption cross section, $\phi$ the quantum yield and $F_\lambda$ the mean spectral actinic flux density inside the chamber. Absorption cross sections and quantum yields were taken from recommendations in literature. The actinic flux spectra within the chamber were calculated in a model using the spectral radiometer measurements as input. As explained in more detail in Bohn et al. (2005) and Bohn and Zilken (2005), this model takes into account chamber specific parameters such as the time-dependent effects of shadings of the chamber steel frame and the transmittance of the Teflon film. RO$_x$ radicals (OH, HO$_2$, RO$_2$) were measured by laser-induced fluorescence (LIF), in which OH is excited at 308 nm (Holland et al., 1995; Fuchs et al., 2011). HO$_2$ and RO$_2$ are chemically converted to OH, so that the sum of radicals could be detected as OH in separate measurement cells (Fuchs et al., 2008, 2011). In the experiments in 2019, HO$_2$ was additionally measured by Br-CIMS as described by Albrecht et al. (2019). The measurements usually agreed within 15 %. The detection sensitivity for RO$_2$ from $\Delta^3$-carene was found to be in the range of the detection sensitivity for methyl peroxy radicals. For some experiments in this work, the fraction of RO$_2$ formed from $\Delta^3$-carene (RO$_{2,\text{carene}}$) is used and calculated as described in the following section. In all experiments, OH reactivity (k$_{OH}$) was measured using a pump-probe instrument (Lou et al., 2010; Fuchs et al., 2017b). An overview of the instrumentation including their accuracies is given in the Supplement (Table S1).



### 2.3 OH reactivity and peroxy radical distribution

The OH reactivity measured in the SAPHIR chamber represents the sum of all species that react with OH. It can be separated into a fraction attributed to inorganic species (NO, CO, NO$_2$) and formaldehyde (here named kOH$_{inorg}$), and a fraction contributed by VOC species (kOH$_{VOC}$). This allows to distinguish between reactions forming RO$_2$ and those that do not. Equation 2 allows to calculate the fraction contributed by VOC by substracting kOH$_{inorg}$ from the total measured reactivity. $k_{OH+X}$ represents the reaction rate constant of the respective compound X with OH.

$$\text{kOH}_{\text{VOC}} = \text{kOH}_{obs} - (k_{\text{OH+NO}}[\text{NO}] + k_{\text{OH+NO2}}[\text{NO}_2] + k_{\text{OH+CO}}[\text{CO}] + k_{\text{OH+HCHO}}[\text{HCHO}]) \tag{2}$$

Included in kOH$_{VOC}$ is the reactivity from $\Delta^3$-carene, but also from reaction products like oxygenated VOCs (OVOCs). kOH$_{carene}$, the fraction of OH-reactivity from $\Delta^3$-carene, can be calculated using its OH reaction rate constant and measured concentrations (Equation 3).

$$\text{kOH}_{\text{carene}} = k_{\text{OH+carene}}[\text{carene}] \tag{3}$$

Since the RO$_2$ measurement is the sum of all RO$_2$ produced in the chamber, it can be assumed that the fraction of RO$_2$ radicals produced by $\Delta^3$-carene to the total RO$_2$ concentration is equal to the ratio of OH-reactivity from the $\Delta^3$-carene + OH reaction to the total measured OH reactivity, assuming that every VOC + OH reaction leads to the formation of an RO$_2$ radical, and that all RO$_2$ species have similar chemical lifetimes. The concentration of RO$_2$ formed by $\Delta^3$-carene oxidation can therefore be estimated using Equation 4.

$$[\text{RO}_{2,\text{carene}}] = \frac{\text{kOH}_{\text{carene}}}{\text{kOH}_{\text{VOC}}}[\text{RO}_{2,\text{obs}}] \tag{4}$$

### 2.4 Determination of reaction rate constants and OH yield from ozonolysis

Reaction rate constants for the reaction of OH with $\Delta^3$-carene and its oxidation product caronaldehyde and for the ozonolysis of $\Delta^3$-carene are determined by minimizing the root mean square error (RMSE) between measured VOC concentration time series and results from a simplified box model while the reaction rate constant is varied. The box model includes a mimimum number

of reactions required to describe the loss of the VOC. For the ozonoylsis and OH oxidation of $\Delta^3$-carene, the model consists of Reactions R1, R2 and the dilution of the trace gases from the chamber replenishment flow. The model for the reaction of caronaldehyde with OH consists of Reactions R3 and R4 and the dilution of trace gases. The model is constrained to measured oxidant concentrations (OH and O$_3$), temperature, pressure and dilution rates. For the determination of the caronaldehyde + OH reaction rate constant, the model is additionally constrained to the measured photolysis frequency.

To determine the OH yield of the ozonolysis of $\Delta^3$-carene, the same method is applied, but the model is constrained to the determined reaction rate constants, while the OH yield is varied. When ozonolysis experiments are conducted in the dark chamber, it is assumed that OH production only occurs through the ozonolysis of $\Delta^3$-carene, and that there are no other photolytic sources (see Section 3.1). For this analysis, it is of no importance whether the correct absolute concentration of $\Delta^3$-carene or caronaldehyde is used in the box model, as the relative decay of modeled and measured $\Delta^3$-carene or caronaldehyde

are compared.



## 2.5 Determination of product yields - organic nitrate RONO$_2$

The yield of nitrates (RONO$_2$) from the reaction of RO$_{2,\text{carene}}$ + NO can be determined from the analysis of the concentrations of reactive nitrogen species in the chamber. NO, NO$_2$ and HONO were directly measured in the experiments and their sum is called NO$_y$* for the analysis in this work (Equation 5):

$$[\text{NO}_y{}^*] = [\text{HONO}] + [\text{NO}_2] + [\text{NO}] \tag{5}$$

The source of all reactive nitrogen species in the experiment is the chamber source of HONO in the sunlit chamber. Its variable source strength Q(HONO) depends on temperature, relative humidity and solar ultraviolet radiation (Rohrer et al., 2005). HONO photolysis leads to the production of NO that is further oxidized to higher nitrogen oxides over the course of the experiment.

As HONO can be reformed by the reaction of NO with OH, a photostationary state between HONO, NO and OH is usually reached within several minutes. Therefore, measurements of NO, OH, $j_{\text{HONO}}$ and HONO can be used to calculate the source strength of HONO (Equation 7).

$$\frac{d[\text{HONO}]}{dt} = Q(\text{HONO}) - j_{\text{HONO}}[\text{HONO}] + k_{\text{OH+NO}}[\text{OH}][\text{NO}] \approx 0 \tag{6}$$

$$Q(\text{HONO}) = j_{\text{HONO}}[\text{HONO}] - k_{\text{OH+NO}}[\text{OH}][\text{NO}] \tag{7}$$

For the experimental conditions in this work ($j_{\text{HONO}} \approx 8 \times 10^{-4}\,\text{s}^{-1}$) photostationarity is reached within 20 minutes. Chemical loss of NO$_y$* in the chamber occurs due to the formation of RONO$_2$ (R6, Table 2) and HNO$_3$ (R2, Table 2). Additionally, NO$_y$* species are lost due to replenishment flow that compensates for chamber leakage and gas sampling of analytical instruments. The difference between the time-integrated production and loss terms can then be used to determine the concentration of NO$_y$* at a given time t.

$$[\text{NO}_y{}^*]_t = \int_t (Q[\text{HONO}] - k_{\text{OH+NO}_2}[\text{OH}][\text{NO}_2] - k_{\text{RO}_2,\text{carene+NO,R10}}[\text{RO}_{2,\text{carene}}][\text{NO}] - L_{dil}))dt' \tag{8}$$

where $L_{dil}$ is the loss due to the replenishment flow, diluting the chamber air with the first order rate coefficient $k_d = 1.6 \times 10^{-5}\,\text{s}^{-1}$. $L_{dil}$ is calculated by Equation 9.

$$L_{dil} = ([\text{NO}] + [\text{NO}_2] + [\text{HONO}])k_d \tag{9}$$

With respect to analysis of total nitrogen oxide concentration in the chamber, Equation 8 assumes that HNO$_3$ and RONO$_2$ formation are effective sinks for NO$_x$ and that it does not play a role, if nitrates remain as HNO$_3$ or RONO$_2$ in the gas-phase or if they are for example deposited on the chamber wall as long as neither NO nor NO$_2$ are reformed. Possible decomposition pathways could e.g. be due to photolysis, which would lead to a reformation of NO$_2$. However, reported atmospheric lifetimes of RONO$_2$ species are in the range of several days due to their small absorption cross sections (Roberts and Fajer, 1989)





much longer than the duration of the chamber experiment. $NO_2$ loss due the formation of nitrate radicals ($NO_3$) from the
reaction of $NO_2$ with ozone is also neglected in Equation 6. For the experimental conditions in this work, the photolytic
backreactions, reforming $NO_2$ from $NO_3$ are fast enough that the $NO_3$ concentration in the sunlit chamber remains negligibly
small. The formation of other oxidized nitrogen species such as acetyl peroxy nitrate (PAN) is also assumed to be negligible.
The thermally unstable PAN species are formed from reaction of acyl peroxy radicals with $NO_2$. For experiments in the
SAPHIR chamber with comparable temperature conditions, the mixing ratios of PAN formed in the oxidation of acetaldehyde
emitted by chamber sources are typically less than $100\,\mathrm{pptv}$. Combining Equations 5 and 8, the amount of $RONO_2$ formed
can be calculated as follows.

$$
\begin{aligned}
[\mathrm{RONO_2}]_t &= \int_t k_{\mathrm{NO+RO_2}(R10)}[\mathrm{RO_{2,carene}}][\mathrm{NO}]dt' \\
&= \phi_{\mathrm{RONO_2}} \times \int_t k_{\mathrm{RO_2+NO}}[\mathrm{RO_{2,carene}}][\mathrm{NO}]dt' \\
&= \int_t (Q[\mathrm{HONO}] - k_{\mathrm{OH \times NO_2}}[\mathrm{OH}][\mathrm{NO_2}] - L_{dil}))dt' - ([\mathrm{NO}] + [\mathrm{NO_2}] + [\mathrm{HONO}])
\end{aligned}
\tag{10}
$$

Applying Equation 10, the reaction yield of organic nitrates ($\phi_{\mathrm{RONO_2}}$) can be derived from the calculated $\Delta[\mathrm{RONO_2}]$ and
measured $\mathrm{RO_{2,carene}}$ concentrations.

To proof this concept of calculating organic nitrate yields, reference experiments with $CH_4$ were performed. An upper limit of
0.001 was found for the nitrate yield. Within the uncertainty of the measurements this value is in good agreement with literature
(Scholtens et al., 1999; Butkovskaya et al., 2012). To exclude possible errors in the analysis for larger molecules for which
nitrate formation is significant, such as unknown chamber sources of reactive nitrate species, a similar analysis was performed
for an $\alpha$-pinene experiment conducted in the SAPHIR chamber. For this experiment, an organic nitrate yield of $(26 \pm 3)\,\%$ was
found, which is in reasonable agreement with the reported literature values. A detailed description of the reference experiments
with $CH_4$ and $\alpha$-pinene will be given in a publication currently in preparation.

## 2.6 Determination of product yields

The experiments conducted in SAPHIR allow to determine the product yields of caronaldehyde from the reaction of $\Delta^3$-carene
with OH and $O_3$. The product yield determination for caronaldehyde is done using measured caronaldehyde concentrations and
relating them to the concentration of $\Delta^3$-carene consumed by OH or $O_3$. The concentrations of $\Delta^3$-carene and caronaldehyde
were measured by VOCUS- or PTR-TOF-MS. A correction was applied to the $\Delta^3$-carene and caronaldehyde concentrations
similar to corrections described by Galloway et al. (2011) and Kaminski et al. (2017). To derive the concentration of $\Delta^3$-carene
that reacted with OH or $O_3$, measured $\Delta^3$-carene concentrations were corrected for dilution in the chamber and the reaction
with $O_3$ or OH, respectively. The measured product concentrations of caronaldehyde were corrected for loss due to photolysis



and dilution.

## 2.7 Determination of radical sources and sinks

The experiments performed in the SAPHIR chamber that investigated the OH oxidation of $\Delta^3$-carene allow to calculate production and destruction rates of the total ROx concentration. The atmospheric lifetimes of the ROx radicals range from only a few seconds for the OH radical to minutes for $HO_2$ and $RO_2$ radicals. Therefore, steady state conditions of ROx concentrations can be assumed, so that radical production and destruction rates are always balanced for the timescale of the chamber experiments. If there are imbalances between the calculated production and destruction rates, chemical reactions leading to the formation or destruction of radicals must be missing in the calculations. Table 2 gives an overview of the formation and loss reactions considered in the analysis including the respective reaction rate constants used. The reaction rate constants of the individual reactions were either taken from recent experimental studies, from measurements in this study or from calculations applying structure-activity relationship (SAR) as described in Jenkin et al. (2019).

The main ROx formation processes include the photolysis of ozone (Reaction R11), HONO (Reaction R12) and HCHO (Reaction R13) as well as the ozonolysis of $\Delta^3$-carene (Reaction R14). The formation of $RO_x$ radicals from the photolysis of caronaldehyde (R4) was also considered. The formation rate of ROx radicals P(ROx) can be calculated by Equation 11.

$$
\begin{aligned}
P(ROx) =\ & j_{HONO}[HONO] + \Phi_{OH-R11} \times j_{O(^1D)}[O_3] + 2 j_{HCHO}[HCHO] \\
& + \Phi_{OH+RO_x} \times k_{14}[\Delta^3carene][O_3] + \Phi_{ROx-R4} \times j_{caronal}[caronal]
\end{aligned}
\tag{11}
$$

$\Phi_X$ indicates the yield of the respective radical X from the given reaction. Loss processes include the reactions of radicals with NOx that lead to the formation of HONO from the reaction of OH with NO (Reaction R5), nitric acid ($HNO_3$) from the reaction of OH with $NO_2$ (Reaction R6) or organic nitrates ($RONO_2$) from the reaction of $RO_{2,carene}$ and NO (Reaction R10). Depending on the experimental conditions, radical loss through radical self-reactions become more important, leading to the formation of hydrogen peroxide ($H_2O_2$) from the reaction of two $HO_2$ radicals (Reaction R7), the formation of peroxides (ROOH) from the reaction of $HO_2$ with $RO_2$ (Reaction R8) and the self-reaction of $RO_2$ (Reaction R9). The loss rate of ROx radicals L(ROx) is calculated by Equation 12.

$$
L(ROx) = (k_6[NO_2] + k_5[NO])[OH] + (k_8[HO_2] + 2 \times k_9[RO_2] + k_{10}[NO])[RO_2] + 2k_7[HO_2]^2
\tag{12}
$$

Direct measurements of all relevant species allow to calculate the total formation and loss rates for the ROx radicals. The error of the loss and production rates of the ROx radicals is determined by error propagation taking uncertainties in the measurements and kinetic parameters into account.





**Table 2.** Formation and loss reactions of the $RO_x$ radicals considered in the budget analysis (Fig. 11). Reaction rate constants are given for 298K and 1 atm. The reaction rate constants in the actual analysis are calculated using temperature and pressure data measured during the experiments in SAPHIR.

| | | reaction | k(298 K, 1 atm) cm³s⁻¹ | reference |
|---|---|---|---|---|
| **radical loss** | R5 | $OH+NO \rightarrow HONO$ | $7.5 \times 10^{-12}$ | Burkholder et al. (2020) |
| | R6 | $OH+NO_2 \rightarrow HNO_3$ | $1.1 \times 10^{-11}$ | Burkholder et al. (2020) |
| | R7 | $HO_2+HO_2 \rightarrow H_2O_2+O_2$ | $1.4 \times 10^{-12}$ | Burkholder et al. (2020) |
| | | $HO_2+HO_2 + M \rightarrow H_2O_2+O_2$ | $1.1 \times 10^{-12\,a}$ | Burkholder et al. (2020) |
| | R8 | $HO_2+RO_{2,obs.} \rightarrow ROOH+O_2$ | $2.3 \times 10^{-11}$ | Jenkin et al. (2019) |
| | R9 | $RO_{2,obs.}+RO_{2,obs.} \rightarrow$ products | $1.3 \times 10^{-13}$ | Jenkin et al. (2019) |
| | R10 | $RO_{2,carene}+NO \rightarrow RONO_2$ | $\Phi^b\,9.0 \times 10^{-12}$ | Jenkin et al. (2019) |
| **radical formation** | R11 | $O_3+h\nu\ (<340nm) \rightarrow O(^1D)+ O_2$ | $j_{O(^1D)}$ | measured |
| | | $O(^1D)+ H_2O \rightarrow 2\,OH$ | $2.1 \times 10^{-10}$ | Atkinson et al. (2004) |
| | | $O(^1D)+ M \rightarrow O(^3P) + M$ | $3.3 \times 10^{-11}$ | Atkinson et al. (2004) |
| | R12 | $HONO +h\nu\ (<340) \rightarrow OH + NO$ | $j_{HONO}$ | measured |
| | R13 | $HCHO+h\nu\ (<335)+ O_2 \rightarrow 2\,HO_2 + CO$ | $j_{HCHO}$ | measured |
| | R14 | $\Delta^3\text{-carene} + O_3 \rightarrow 0.65\times OH + RO_2 + \text{products}$ | $4.4 \times 10^{-17}$ | Table 3, this work |
| | R4 | $\text{caronaldehyde} +h\nu \rightarrow HO_2 + RO_2 + \text{products}$ | $j_{caronal}$ | measured |

$^a$ pressure dependent value given as $4.6 \times 10^{-32} \times [M]$ in Burkholder et al. (2020) $^b$ the yield $\Phi$ of this reaction is determined in Section 3.2.3.

## 3 Results and Discussion

### 3.1 Ozonolysis of $\Delta^3$-carene

The ozonolysis of $\Delta^3$-carene was investigated in the dark SAPHIR chamber in two experiments in order to determine the rate constant (experiment E3) and OH yield (experiment E4) of the ozonolysis reaction, and the yield of caronaldehyde. Measured
275 timeseries of $O_3$, $\Delta^3$-carene and caronaldehyde are shown in Fig. 2. In total, $6.5\,\mathrm{ppbv}$ of $\Delta^3$-carene was consumed in experiment E3, and $7\,\mathrm{ppbv}$ in experiment E4. The roof of the chamber was closed for the whole duration of both experiments, to eliminate photolytical OH production. Since there was no OH scavenger present in experiment E3, the reaction system was also influenced by OH that is formed from ozonolysis (Sect. 3.1.2). OH concentrations were in the range of $1.0$ to $2.0 \times 10^6\,\mathrm{cm}^{-3}$. CO was injected into the chamber as an OH scavenger prior to the beginning of experiment E4.

280





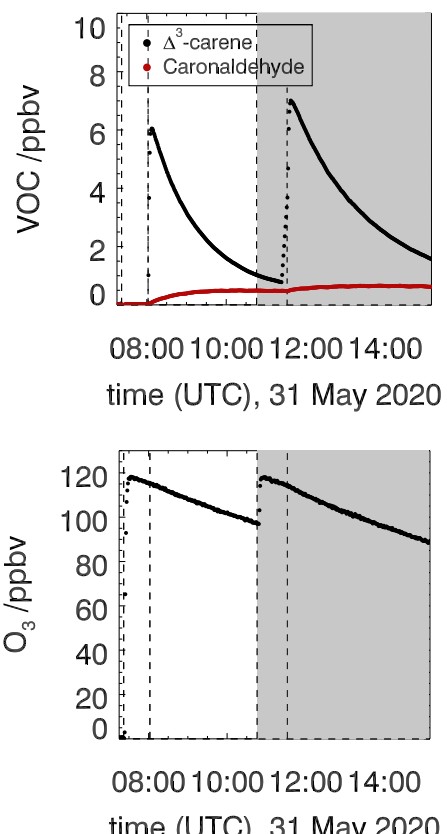

**Figure 2.** Measured $\Delta^3$-carene, caronaldehyde, OH, O$_3$ and CO concentrations in the experiment investigating the ozonolysis of $\Delta^3$-carene (experiment E3). The shaded area indicates the time where CO was injected as an OH scaveger (experiment E4). Vertical lines give the times, when trace gases were injected.

### 3.1.1 Rate constant of the ozonolysis reaction of $\Delta^3$-carene

Optimization of the ozonolysis reaction rate constants as described in Sect. 2.4 results in a value of $(4.4 \pm 0.2) \times 10^{-17}\,\mathrm{cm^3 s^{-1}}$. The stated error arises from the accuracy of the O$_3$ and $\Delta^3$-carene measurements. The average temperature inside the SAPHIR chamber during the experiment was $300\,\mathrm{K}$. The value reported in this study is slightly higher than values reported by Atkinson et al. (1990) and Chen et al. (2015), but still agrees within the stated errors (Table 3).

Reaction rate constants determined by Atkinson et al. (1990) and Chen et al. (2015) were $(4.05 \pm 0.4) \times 10^{-17}\mathrm{cm^3 s^{-1}}$ and $(3.7 \pm 0.4) \times 10^{-17}\mathrm{cm^3 s^{-1}}$, respectively. Both values were obtained from relative reaction rate measurements by comparing the rate of decay of $\Delta^3$-carene from ozonolysis to the rate of decay of a well-known reference compound ($\alpha$-pinene in Atkinson



**Table 3.** Rate constants of the reaction of $\Delta^3$-carene with $O_3$ and OH and of the reaction of caronaldehyde with OH. Experimental values obtained in this study are compared to reported literature values.

| Reaction | Reaction rate constant / $\mathrm{cm^3s^{-1}}$ | Technique | Temperature / K | Reference |
|---|---|---|---|---|
| $\Delta^3$-carene + $O_3$ | $(4.4\pm0.2)\times10^{-17}$ | absolute rate | 300 | this study |
| | $(5.9\pm1)\times10^{-17}$ | relative rate | 295 | Witter et al. (2002) |
| | $(3.7\pm0.2)\times10^{-17}$ | relative rate | 299 | Chen et al. (2015) |
| | $(4.9\pm0.8)\times10^{-17}$ | absolute rate[b] | 299 | Chen et al. (2015) |
| | $(3.5\pm0.2)\times10^{-17}$ | absolute rate[c] | 299 | Chen et al. (2015) |
| | $(3.7\pm0.2)\times10^{-17}$ | absolute rate[d] | 299 | Chen et al. (2015) |
| | $(3.8\pm0.2)\times10^{-17}$ | relative rate | 296 | Atkinson et al. (1990) |
| | $(5.2\pm0.6)\times10^{-17}$ | absolute rate | 296 | Atkinson et al. (1990) |
| | $4.8\times10^{-17}$ | SAR[e] | 298 | Jenkin et al. (2020) |
| $\Delta^3$-carene + OH | $(8.0\pm0.5)\times10^{-11}$ | absolute rate | 304 | this study |
| | $(8.0\pm0.1)\times10^{-11a}$ | relative rate | 304 | Dillon et al. (2017) |
| | $(8.7\pm0.4)\times10^{-11}$ | relative rate | 294 | Atkinson et al. (1986) |
| | $8.5\times10^{-11}$ | SAR | 298 | Peeters et al. (2007) |
| caronaldehyde + OH | $(4.1\pm0.3)\times10^{-11}$ | absolute rate | 300 | this study |
| | $(4.8\pm0.8)\times10^{-11}$ | relative rate | 296 | Alvarado et al. (1998) |
| | $(12.1\pm0.8)\times10^{-11}$ | relative rate | 298 | Hallquist et al. (1997) |
| | $(2.9\pm0.8)\times10^{-11}$ | SAR | 298 | Jenkin et al. (2018) |

[a] temperature dependent reaction rate coefficient given by Dillon et al. (2017) : $(2.48\pm0.14)\times\exp{(357\pm17)}/T\times10^{-17}\,\mathrm{cm^3s^{-1}}$

[b] measured in a flow reactor [c] measured in a $7.3\,\mathrm{m^3}$ Teflon chamber [d] measured in a $90\,\mathrm{m^3}$ Teflon chamber [d] structure activity relationship

et al. (1990), cyclohexene in Chen et al. (2015)). Additionally both studies also determined the reaction rate constant with an absolute rate technique using a pseudo-first order approach. In Atkinson et al. (1990), reactive impurities in the used $\Delta^3$-carene sample were reported, whose presence would reduce the initially measured reaction rate constant from the absolute technique from $(5.2\pm0.6)\times10^{-17}\,\mathrm{cm^3s^{-1}}$ to $(4.1\pm0.6)\times10^{-17}\,\mathrm{cm^3s^{-1}}$. Chen et al. (2015) used three different set-ups to determine the reaction rate constant with an absolute rate technique: two simulation chambers with volumes of $90\,\mathrm{m^3}$ (HELIOS) and $7.3\mathrm{m^3}$ and a laminar flow reactor. The values determined in the simulation chambers were $(3.5\pm0.2)\times10^{-17}\,\mathrm{cm^3s^{-1}}$ in the smaller simulation chamber and $(3.7\pm0.2)\times10^{-17}\,\mathrm{cm^3s^{-1}}$ in HELIOS. A rate constant of $(4.9\pm0.8)\times10^{-17}\,\mathrm{cm^3s^{-1}}$ was determined in the flow reactor. The $30\,\%$ discrepancy between the determined reaction rate constant is explained by interferences in the $O_3$ measurements and increased uncertainties due to higher wall losses in the flow reactor. Witter et al. (2002) reported a faster reaction rate constant of $(5.9\pm1.0)\times10^{-17}\,\mathrm{cm^3s^{-1}}$, obtained with a relative rate technique using $2-\mathrm{methyl}-2-\mathrm{butene}$ as reference compound. The value reported by Witter et al. (2002) is higher than the other reported values. Although the reason



for this discrepancy is not clear, in their study Witter et al. (2002) also determined the ozonolysis reaction rate constants for other monoterpenes, e.g. limonene, $\alpha$-terpinene and $\alpha$-pinene, and the determined values were mostly on the upper end of reported literature values. The rate constants were on average $20\,\%$ higher than comparable values, which may point to a

general overestimation of monoterpene reaction rate constants. The IUPAC Task Group on Atmospheric Chemical Kinetic Data Evaluation recommend to use a value of $(4.9 \pm 0.2 \times 10^{-17})\,\mathrm{cm^3 s^{-1}}$ (Atkinson et al., 2004), which is an average of the relative rate constants determined by Atkinson et al. (1990) and Witter et al. (2002) and is in good agreement with the value determined in this work. Using the SAR published by Jenkin et al. (2020) a reaction rate constant of $4.7 \times 10^{-17}\,\mathrm{cm^3 s^{-1}}$ can be calculated. The value reported in this study is in relatively good agreement with this theory derived value.

### 310    3.1.2    Determination of the OH yield from the ozonolysis of $\Delta^3$-carene

During experiment E3, when no OH scavenger was present, $\Delta^3$-carene was not only consumed by $O_3$, but also by OH that is produced from the ozonolysis reaction. Because OH production from the ozonolysis was the only OH source, the OH yield from ozonolysis of $\Delta^3$-carene can be determined from this experiment. The chamber roof was closed for the whole experiment and there are no photolytic sources for OH formation. The reaction rate constant determined in the previous section for the

period, when an OH scavenger was present in the SAPHIR chamber, is used in the box model to determine the OH yield of the ozonolysis as explained in Sect. 2.4. The OH yield from ozonolysis is optimized until the measured decay of $\Delta^3$-carene matches the modeled decay (Fig. 3).

The OH yield is found to be $0.65 \pm 0.10$ in the experiment in this work. The error is mainly due to the uncertainties in

the reaction rate constants and the accuracies of the $O_3$ measurement. Reported literature values for the OH yield from the ozonolysis of $\Delta^3$-carene determined in experiments are $1.1 \pm 1.0$ (Atkinson et al., 1992) and $0.86 \pm 0.11$ (Aschmann et al., 2002a). Wang et al. (2019) theoretically calculated the OH yield to be 0.56 to 0.59. Table 4 shows the value determined here compared with reported literature values. The value reported here is closer to the theoretical than to the other measured values, but agrees with the yield reported by Aschmann et al. (2002a) within the reported uncertainties. As discussed in Wang et al.

(2019), the high OH yield reported in Atkinson et al. (1992) could have been overestimated because of the high uncertainty in the determination of the OH yield. Atkinson et al. (1992) derived the OH yield by measuring the concentration of cyclohexanol and cyclohexanone produced in the reaction of OH and cyclohexane that was used as OH scavenger in their experiments. More recent investigations of the yield of cyclohexanol and cyclohexanone from the reaction of cyclohexane and OH indicate a larger of yield, 0.88 (Berndt et al., 2003), as compared to the value of 0.5 used by Atkinson et al. (1992). This higher value

would reduce the OH yield in the ozonolysis of $\Delta^3$-carene in the experiments by Atkinson et al. (1992) to 0.6, which would agree well with the OH yield determined in this work and the value calculated by Wang et al. (2019).

The ozonolysis of $\Delta^3$-carene is initiated by $O_3$ attacking the C-C double bond, forming an energy-rich primary ozonide (POZ). Mostly, the energy retained in the POZ leads to a decomposition into Criegee intermediates, retaining one structure bearing

a carbonyl functionality on one side of the molecule and the Criegee funcionality on the other. These Criegee intermediates

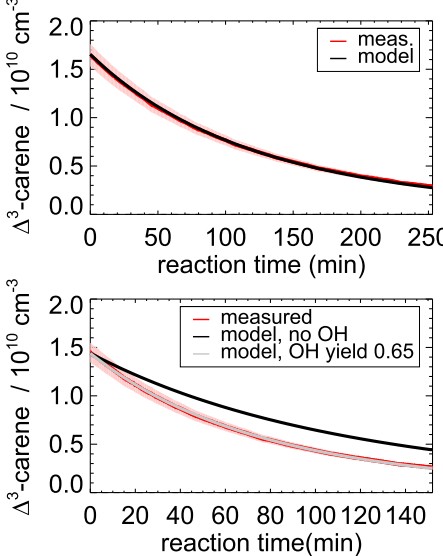

**Figure 3.** Measured and modeled $\Delta^3$-carene concentrations for the ozonolysis experiment E4. During the second part of the experiment (upper panel), 90 ppmv of an OH scavenger (CO) were injected to suppress OH, so that $\Delta^3$-carene reacted only with ozone. The modeled decay is fitted with a rate constant of $(4.4 \pm 0.2) \times 10^{-17} \, \mathrm{cm^3 s^{-1}}$. During the first part of the experiment (lower panel), when no CO was present, the measured decay of $\Delta^3$-carene is signficantly faster than expected from ozonolysis alone. The measured time series of $\Delta^3$-carene can be best matched, if an OH yield of $0.65 \pm 0.10$ from its ozonolysis reaction is assumed. Red colored areas indicate the accuracy of the measured $\Delta^3$-carene concentrations.

**Table 4.** OH yield of the reaction of $\Delta^3$-carene with $O_3$. Experimental values obtained in this study are compared to reported literature values.

| OH yield | Technique | Reference |
|---|---|---|
| $(0.65 \pm 0.10)$ | absolute rate | this study |
| $(1.1 \pm 1.0)$ | OH scavenging with $c-$hexane | Atkinson et al. (1992) |
| $(0.6 \pm 1.0)$ | OH scavenging with $c-$hexane[a] | Atkinson et al. (1992) |
| $(0.86 \pm 0.11)$ | OH scavenging with $2-$butanol | Aschmann et al. (2002b) |
| 0.56 to 0.59 | theoretical calculations | Wang et al. (2019) |

[a] with revistited yields of $c-$hexanone and $c-$hexanol by Berndt et al. (2003) as explained in the text

then isomerize to form dioxiranes, secondary ozonides (SOZ) or vinyl hydroperoxides. A theoretical study of the ozonolysis of $\Delta^3$-carene by Wang et al. (2019) found the formation yields dioxiranes, SOZ and vinyl hydroperoxides to be 0.16, 0.24 and 0.56, respectively. Collisional stabilization of the Criegee intermediates was found to be of minor importance with a yield of





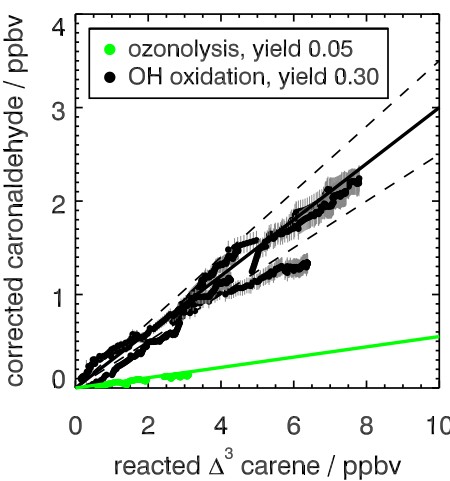

**Figure 4.** Reacted $\Delta^3$-carene plotted against the corrected caronaldehyde concentration for the OH oxidation and ozonolysis experiments. Concentrations were corrected for dilution and chemical loss (see text for details). Caronaldehyde yields from the oxidation of $\Delta^3$-carene are determined as the slopes of the shown linear regression lines (black: OH oxidation, green: ozonolysis).

only 0.04 in the same study. OH (and an alkoxy radical) is formed from the vinyl hydroperoxide (ROOH) by breakage of the
$-O-OH$ bond (Aschmann et al., 2002a; Ma et al., 2009). The fate of the alxoxy radical formed from this reaction has yet to be investigated, but it might result in the formation of highly oxidized molecules (HOM) (Wang et al., 2019).

### 3.1.3 Caronaldehyde yield from ozonolysis

The caronaldehyde yield for the $\Delta^3$-carene + O$_3$ reaction was determined from experiment E4, where an OH scavenger was
injected into the chamber, so that caronaldehyde is exclusively formed form the ozonolysis reaction. The corrections described in Sect. 2.5 were applied. Caronaldehyde was formed from the ozonolysis of $\Delta^3$-carene with a yield of $(5.5 \pm 2)\,\%$ as shown in Fig. 4. The uncertainty is derived from measurements and errors of the applied corrections (Section 2.6) for the respective experiment.

The determined caronaldehyde yields from this study and reported literature values are given in Table 5. The obtained value is in reasonably good agreement with most of the reported literature values. Yu et al. (1999) reported a caronaldehyde yield of $8\,\%$, using $2-$butanol to scavenge OH radicals. Hakola et al. (1994) determined the caronaldehyde yield to be $\leq 8\,\%$. In the presence of cyclohexane as an OH scavenger, caronaldehyde was not detected in the ozonolysis of $\Delta^3$-carene in the experiments conducted by Hakola et al. (1994). Ma et al. (2009) reported a caronaldehyde yield of $(0.47 \pm 0.05)\,\%$ from filter
samples, using cyclohexane, methanol or ethanol to scavenge OH radicals. However, due to its relatively high vapour pressure,





**Table 5.** Caronaldehyde yields for the reaction of $\Delta^3$-carene with OH and OH and caronaldehyde yields for the reaction of $\Delta^3$-carene with $O_3$ obtained from this study compared to literature values. Given $\Delta^3$-carene concentrations are inital concentrations.

| Reaction | Yield / % | Exp. conditions | | | Reference | |
|---|---|---|---|---|---|---|
| | caronaldehyde | $\Delta^3$-carene /ppbv | NO /ppbv | RH / % | product quantification | |
| $\Delta^3$-carene + OH | $30 \pm 5$ | 5 | 0.8 | 45 | PTR-MS | this study |
| | 31 | 975 | 9756 | 0 | GC-FID[a] | Arey et al. (1990) |
| | $34 \pm 0.8$ | 9800 | 0 | 0 | GC-FID | Hakola et al. (1994) |
| $\Delta^3$-carene + $O_3$ | $5.5 \pm 2$ | 6 | 0.1 | 18 | PTR-MS | this study |
| | $8.5^b$ | 89.9 | n/a[c] | n/a[c] | GC-MS | Yu et al. (1999) |
| | $\leq 8^d$ | 9800 | 0 | 0 | GC-FID/GC-MS | Hakola et al. (1994) |
| | $0.47 \pm 0.05^e$ | 15000 | 0 | 0 | GC-FID/GC-MS | Ma et al. (2009) |

[a] GC-FID: gas chromatograph with flame ionization detector, GC-MS: gas chromatograph with mass spectrometer. [b] combined yield for gas and aerosol phase. [c] no values given in the publication. [d] yield obtained in the absence of an OH scavenger. [e] obtained from filter samples.

it is likely that caronaldehyde was mainly present in the gas phase, so that only a small fraction of the formed caronaldehyde was collected on the filters.

The formation of caronaldehyde from $\Delta^3$-carene ozonolysis most likely results from the stabilization and subsequent reaction with water of one of the Criegee intermediates, the formation mechanism of caronaldehyde in the absence of water has yet to be clarified (Ma et al., 2009). The stabilization of the Criegee intermediate is found to be a minor pathway by Wang et al. (2019), possibly explaining the small caronaldehyde yield found for the ozonolysis of $\Delta^3$-carene. Further reaction products that have not been measured in this study include a range of multifunctional organic acids according to studies conducted by e.g. Ma et al. (2009).

## 3.2 OH reaction of $\Delta^3$-carene

The first oxidation steps of the OH-induced photochemical oxidation of $\Delta^3$-carene relevant for this study are shown in Fig. 5 (Colville and Griffin, 2004). The OH oxidation is initiated by addition of OH to the C-C double bond or by the abstraction of an H-atom rapidly followed by addition of $O_2$. From the OH addition two peroxy radical isomers are formed. The branching ratios of the specific attack on the C-C double bond (Fig. 5) and the H-abstraction are estimated using structure-activity rela-
tionship (SAR) method described in Jenkin et al. (2018). Branching ratios are 10 % for H-abstraction, 30 % for OH addition to C1 and 60 % for for OH addition to C2 of the double bond. Due to the small fraction of H-abstraction, a relatively small influence of these reactions on the results of the following analysis can be expected.







**Figure 5.** Simplified scheme of the first reaction steps of the OH photooxidation of $\Delta^3$-carene (adapted from Colville and Griffin (2004). Yields shown in black are from SAR by Jenkin et al. (2018). H-abstraction has only little influence on the presented product yields. Additional pathways for the reaction of $RO_2$ with $HO_2$ or $RO_2$ with $RO_2$ are not shown because the $RO_2 + RO_2$ and $RO_2 + HO_2$ contributed only $\leq 35\,\%$ to the fate of $RO_2$.

Peroxy radicals ($RO_2$) can react with NO, $HO_2$ or undergo radical self-reactions with $RO_2$. The reaction of $\Delta^3$-carene
derived $RO_{2,carene}$ with NO and possible reaction products are discussed in Sect. 3.2.2 and 3.2.3. $RO_2 + RO_2$ and $RO_2 +$ $HO_2$ reactions are discussed in Sect. 3.2.4.

OH induced photooxidation of $\Delta^3$-carene was investigated in the SAPHIR chamber during two experiments with NOx mixing ratios below 1 ppbv (Table 1). Figure 6 shows an overview of all measured species in experiment E1 that are representative for experiments at low NOx mixing ratios. Time series of concentration measurements for the other experiment is shown in the
Supplement (Fig. S1). For all experiments, $\Delta^3$-carene was injected three times into the chamber, increasing the mixing ratio by approximately 5 ppbv each injection.





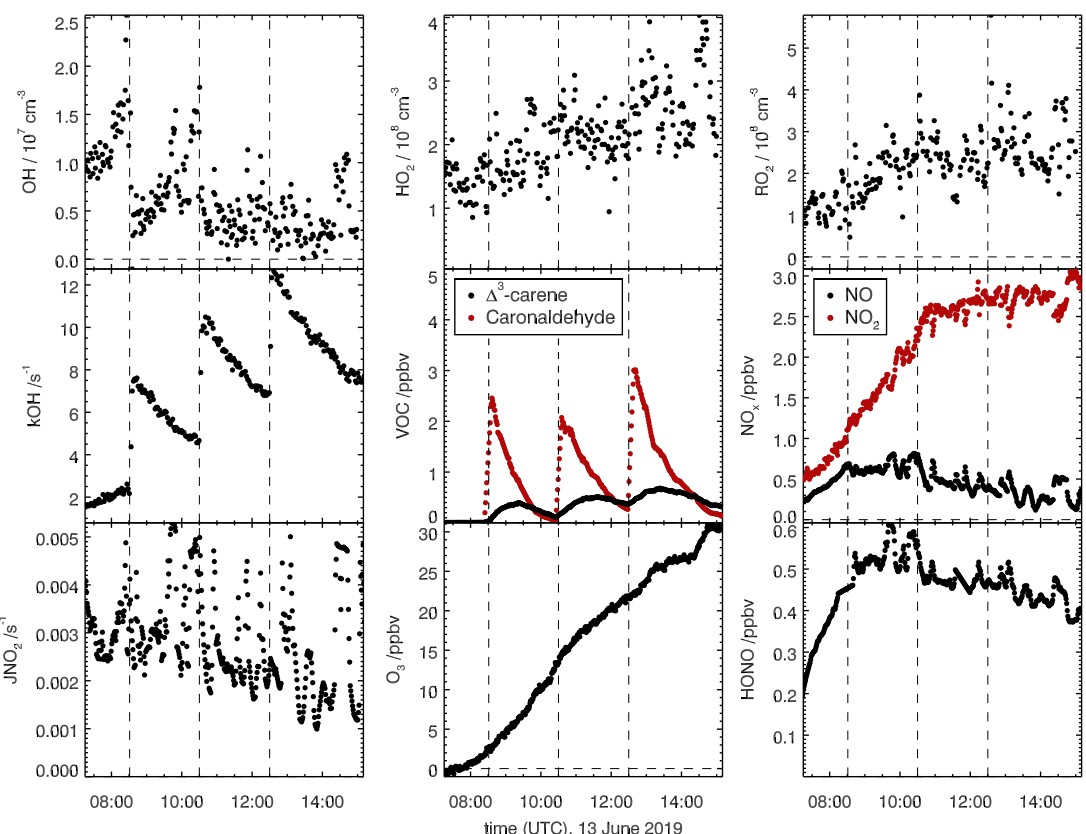

**Figure 6.** Overview of measured concentrations for selected species in the SAPHIR chamber for the $\Delta^3$-carene + OH oxidation experiment E1. Dashed lines indicate times when $\Delta^3$-carene was injected into the chamber. Measured concentrations are shown in the illuminated chamber only starting with the opening of the roof ('zero-air' phase).

### 3.2.1 Rate constant of the OH + $\Delta^3$-carene reaction

In order to determine the rate constant of the reaction of $\Delta^3$-carene with OH, measured time series were compared to model results from a box model, in which OH and $O_3$ concentrations were constrained to the measurements. A reaction rate constant

was determined from all experiments. The determined value represents the mean value from all experiments. On average, the temperature inside the SAPHIR chamber was $304\,\mathrm{K}$ during the experiments. OH concentrations usually ranged from 5 to $8 \times 10^6\,\mathrm{cm}^{-3}$. Ozonolysis only played a minor role in the experiments contributing to a maximum of $5\,\%$ to the $\Delta^3$-carene consumption. The reaction rate constant for the ozonolysis reaction was taken from this work (Section 3.1.1).

The optimized rate constant of the OH reaction with $\Delta^3$-carene is $(8.0 \pm 0.5) \times 10^{-11}\,\mathrm{cm^3 s^{-1}}$ as shown in Fig. 7.

Table 3 shows a comparison of reported literature values to the reaction rate constants obtained in this study. The determined value for the OH + $\Delta^3$-carene reaction rate constant from this study agrees well with both experimentally and

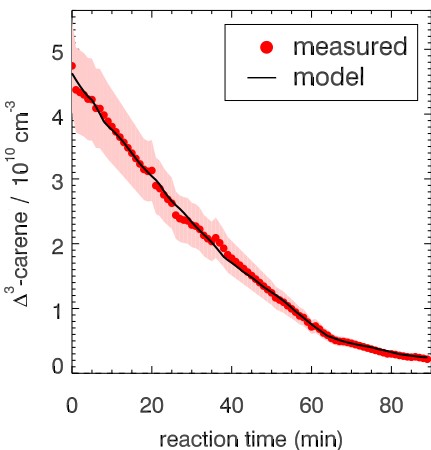

**Figure 7.** Comparison of the modeled and measured $\Delta^3$-carene decay for the OH-oxidation experiment E2. Shown is the result of optimization of the rate constant of the OH reaction with $\Delta^3$-carene in the model calculations resulting in a reaction rate constant of $7.9 \times 10^{-11} \text{cm}^3\text{s}^{-1}$.

theoretically derived literature values. Dillon et al. (2017) determined a temperature-dependent rate coefficient, for comparability with data in this work, only the rate coefficient determined at $304\,\text{K}$ is used. Dillon et al. (2017) determined a reaction rate constant of $(8.1 \pm 0.1) \times 10^{-11}\,\text{cm}^3\text{s}^{-1}$ using an absolute rate approach and Atkinson et al. (1986) reported a value of
$(8.7 \pm 0.4) \times 10^{-11}\,\text{cm}^3\text{s}^{-1}$ at $294\,\text{K}$ using a relative rate determination approach. From a site specific structure-activity relationship (SAR) Peeters et al. (2007) predict the reaction rate constant to be $8.5 \times 10^{-11}\,\text{cm}^3\text{s}^{-1}$ at $298\,\text{K}$. In this reaction rate constant, the contribution of H-abstraction is not considered.

### 3.2.2 Caronaldehyde yield from the reaction of $\Delta^3$-carene with OH

Table 5 compares the product yields of the $\Delta^3$-carene + OH reaction obtained in this study to reported literature values. The caronaldehyde yield of $(30 \pm 5)\,\%$ determined in this study as shown in Fig. 4 in good agreement with reported literature values within the range of the uncertainty of the measurements. The given error reflects the range of values determined in the considered experiments. Arey et al. (1990) reported a yield of $31\,\%$ and Hakola et al. (1994) a yield of $(34 \pm 0.8)\,\%$. In both studies, caronaldehyde was quantified using GC-FID. Hakola et al. (1994) additionally used GC-MS and [1]H-NMR to verify
the structure and purity of the measured compound.

As shown in Fig. 5, caronaldehyde is mainly formed from the decomposition of alkoxy radicals (RO). These alkoxy radicals are mainly formed from the reaction of $RO_2$ with NO. A similar RO radical is also formed from the $RO_2 + RO_2$ reaction, and from the photolysis of hydroperoxides (ROOH) that result from the $RO_2 + HO_2$ reaction. Since the $RO_2 + NO$ reaction mainly leads to the formation of RO (the branching ratio of an alternative pathway is discussed in Sect. 3.2.3), yields of





caronaldehyde as found in this and previous studies can be expected. Other reaction products of the $\Delta^3$-carene + OH reaction determined in previous studies include formaldehyde with a yield of $20\%$ (Orlando et al., 2000) and acetone with a yield of $15\%$ (Reissell et al., 1999; Orlando et al., 2000). Caronic acid, hydroxy-caronic acid isomers and hydroxy-caronaldehyde isomers have additionally been found in the aerosol phase of smog chamber experiments investigating the $\Delta^3$-carene + OH reaction by Larsen et al. (2001).

### 415  3.2.3  Determination of alkyl nitrate yield for the reaction of $\Delta^3$-carene + OH

Organic nitrates are formed from the reaction of $RO_2$ radicals with NO as shown in Fig. 5 for the reaction of the $RO_2$ formed in the first oxidation step of $\Delta^3$-carene. An alternative pathway for the NO + $RO_2$ reaction is the formation of an alkoxy radical and $NO_2$ that ultimately leads to the formation of caronaldehyde (Section 3.2.2). The nitrate yield $\Phi RONO_2$ for the reaction of $\Delta^3$-carene + OH is determined following the procedure described in Sect. 2.4 using $RO_{2,carene}$. Figure 8 shows
the accumulation of reactive NOy species over the course of the experiment E2. In total, $8.1\,ppbv$ of reactive nitrogen species were formed over the course of the experiment. The contributions of NO, $NO_2$ and HONO are almost constant with $2.2\,ppbv$, while the contributions of $RONO_2$ and $HNO_3$ increase continuously over the course of the experiments due to their continuous production from the reaction of $RO_{2,carene}$ with NO and OH with $NO_2$, respectively. The formation of $HNO_3$ and $RONO_2$ before the injection of $\Delta^3$-carene is not relevant for the analysis and therefore not shown in Fig. 8.


An organic nitrate yield of $(25\pm4)\%$ was found from experiments E1 and E2. To our knowledge, only one other study investigated the organic nitrate yield of $\Delta^3$-carene. Based on $RONO_2$ measurements in the aerosol phase using a TD-LIF instrument (thermal dissosciation laser induced fluorescence), Rollins et al. (2010) calculated an organic nitrate yield of $25\%$ for the photooxidation of $\Delta^3$-carene in experiments with high NOx conditions ($100\,ppbv$ of $\Delta^3$-carene and $500\,ppbv$ of NO). This
yield represents the fraction of SOA molecules that are hydoxy-nitrates and Rollins et al. (2010) suggest that the organic nitrate fraction of SOA molecules produced in photooxidation are similar to the total yield of $RONO_2$ from $RO_2$ + NO. The SAR by Jenkin et al. (2019) predicts an organic nitrate yield of $19\%$. This calculation is based on the number of C-atoms in the peroxy radical, and therefore possibly has a relatively high uncertainty. The value obtained in this study for experiments with $NO_x$ mixing ratios below $1\,ppbv$ is in good agreement with both the experimental and the SAR derived values. Due to their
structural similarities, it can be assumed that organic nitrate yields for $\Delta^3$-carene could be comparable to reported determined organic nitrate yields of $\alpha$-pinene. Noziere et al. (1999) report an organic nitrate yield of $(18\pm9)\%$ and Rindelaub et al. (2015) $(26\pm7)\%$ for the reaction of $\alpha$-pinene with OH. The organic nitrate yield of $1\%$ by Aschmann et al. (2002b) was not measured directly, but approximated from API-MS measurements. This leads to a high uncertainty, and likely explains the difference to the values by Noziere et al. (1999); Rindelaub et al. (2015). The nitrate yield obtained in this study for $\Delta^3$-carene
is in reasonable agreement with the values reported for $\alpha$-pinene, as can be expected due to their structural similarities.


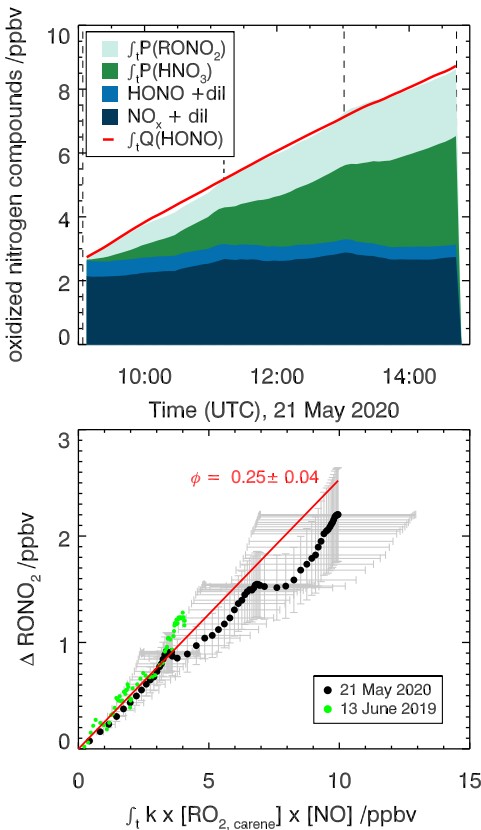

**Figure 8.** Determination of the organic nitrate yield for the $\Delta^3$-carene + OH oxidation experiment E2. The red line shows teh time integrated HONO emission calculated as described in the text. $NO_x$ and HONO concentrations were measured, while the time integrated $NO_2$ loss by $HNO_3$ formation and time integrated NO loss by $RONO_2$ formation are calculated as described in the text. The yield of $(25 \pm 4)\,\%$ for alkyl nitrates is optimized such that the total NOy produced over the course of both experiments E1 and E2 is accounted for.

### 3.3    Photooxidation of caronaldehyde

A simplified reaction scheme for caronaldehyde degradation chemistry is shown in Fig. 9. Caronaldehyde can be oxidized by the reaction with OH, forming $RO_2$ radicals through H-abstraction and fast subsequent addition of $O_2$. The formed peroxy
radicals can then undergo reactions with NO, $HO_2$ and $RO_2$, resulting in similar product species such as explained before. Photolysis of caronaldehyde is an additional loss path. Due to their structural similarities, it can be assumed that caronaldehyde photolyses in a similar way like pinonaldehyde. Photolysis leads to the C-C bond scisson next to the aldehydic functional unit of the molecule, leading to the formation of HCO and an alkyl radical. Both species subsequently react with $O_2$, forming CO, $HO_2$ and a peroxy radical.






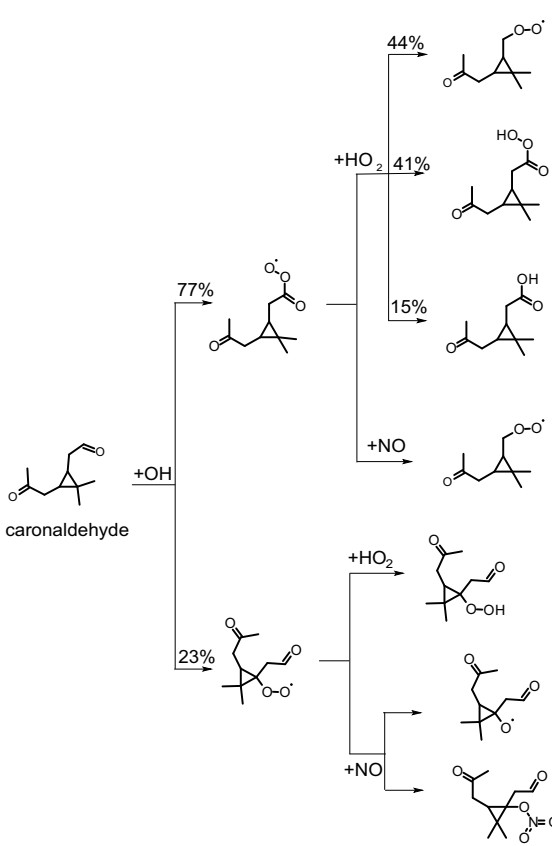

**Figure 9.** Simplified scheme of the first reaction steps of the OH photooxidation of caronaldehyde, adapted from the oxidation mechanism of pinonaldehyde. Yields shown in black are from SAR by Jenkin et al. (2018). Additional reactions like e.g. the formation of carbonic acids from $RO_2$ with $HO_2$ are not shown in the scheme.

The reaction of caronaldehyde with OH was investigated in experiment E6. Caronaldehyde was injected twice into the chamber to reach mixing ratios of 5 ppbv directly after the first and 6 ppbv directly after the the second injection. The OH concentration was $7 \times 10^6 \, \mathrm{cm}^{-3}$ for the period after the first injection. Prior the second injection, 50 ppmv of CO was injected into the chamber as an OH scavenger to study the photolysis of caronaldehyde separately. The temperature in the chamber ranged from
295 K to 305 K during the experiment.

### 3.3.1 Photolysis frequency of caronaldehyde

As CO was injected into the chamber as an OH scavenger and wall loss was found to be negligible in the timeframe of the experiment, the decay of caronaldehyde measured can only be due to photolysis and dilution. The absorption spectrum of caronaldehyde was measured by Hallquist et al. (1997) in the range from 275 to 340 nm at 300 K. This spectrum and
an assumed quantum yield for dissociation of one was used to calculate the photolysis frequency for the conditions of the



experiment in the sunlit chamber using Equation 1 (Section 2.2) resulting in a mean loss of caronaldehyde due to photolysis of $1.3 \times 10^{-5}\,\mathrm{s}^{-1}$. The calculated photolysis frequency was included in a model to compare the measured to the modeled decay. The modeled decay was found to be too slow and requires a photolysis frequency that is larger by a factor of 7 to match the experimental decay. The absorption cross section by Hallquist et al. (1997) used in this work is the only reported measurement.

It was measured in a $0.48\,\mathrm{m}^3$ borosilica reactor at low pressure. During their experiments, dichlormethane and methanol solvents used in the synthesis of caronaldehyde were present in the reactor. Even though the authors corrected for possible interferences caused by these two species regarding the caronaldehyde concentration measured by IR spectroscopy, potential errors in the measurent of caronaldehyde concentrations might explain the discrepancy. Although there is no mechanistic explanation, it cannot be fully excluded that the faster decay observed in the experiments in this work compared to that by

Hallquist et al. (1997) is the result of an alternative OH independant loss process occuring in the illuminated chamber. The absorption cross sections that have to be applied to explain the decay of caronaldehyde in photolysis experiments in SAPHIR suggest further investigation of this reaction and the absorption cross section of caronaldehyde.

### 3.3.2   Rate constant of the caronaldehyde + OH reaction

The measured decay of caronaldehyde was compared to the results from the box model constraining the photolysis frequency

to the value with a correction by a factor of 7 calculated as explained above. The temperature inside the SAPHIR chamber was $300\,\mathrm{K}$, the OH concentration was $8 \times 10^6\,\mathrm{cm}^{-3}$. The optimum OH reaction rate constant is $(3.6 \pm 0.7) \times 10^{-11}\,\mathrm{cm}^3\mathrm{s}^{-1}$. The reaction rate constant was also optimized with the model constrained to the photolysis frequency as calculated with the absorption spectrum measured by Hallquist and a quantum yield of 1. This yields an OH reaction rate constant of $(5.5 \pm 0.7) \times 10^{-11}\,\mathrm{cm}^3\mathrm{s}^{-1}$. Both values agree reasonably well with the reaction rate constant of $(4.8 \pm 0.8) \times 10^{-11}\,\mathrm{cm}^3\mathrm{s}^{-1}$ measured

by Alvarado et al. (1998) within the error. Structure activity relationship by Jenkin et al. (2018) predicts a total reaction rate constant of $2.9 \times 10^{-11}\,\mathrm{cm}^3\mathrm{s}^{-1}$ which is consistent with the two measured values within the uncertainty of SAR. Hallquist et al. (1997) also determined the reaction rate constant of caronaldehyde with OH resulting in a value of $(12.1 \pm 3.6) \times 10^{-11}\,\mathrm{cm}^3\mathrm{s}^{-1}$. This value is significantly higher than both values reported here and by Alvarado et al. (1998). Both Alvarado et al. (1998) and Hallquist et al. (1997) determined the reaction rate constant using a relative rate technique and both studies were

performed in smaller reaction volumes than the $270\,\mathrm{m}^3$ SAPHIR chamber. The measurements by Hallquist et al. (1997) were performed in a $0.153\,\mathrm{m}^3$ borosilica glass reactor, while the studies by Alvarado et al. (1998) were performed in an $7.9\,\mathrm{m}^3$ Teflon chamber. Wall losses of caronaldehyde in the range of $(4-7) \times 10^{-5}\,\mathrm{s}^{-1}$ were observed in the borosilica chamber in the dark and further increased when the chamber was irradiated. Even though the determined reaction rate constant is corrected for the measured wall loss, it causes further uncertainty. Additionally, the initial caronaldehyde mixing ratio in the experiments

in this work and the study performed by Alvarado et al. (1998) were significantly lower than those used by Hallquist et al. (1997) (20 ppbv (this study), 113 ppbv (Alvarado et al., 1998), 3252 ppbv (Hallquist et al., 1997), respectively). The reason for the high reaction rate constant reported by Hallquist et al. (1997) is not entirely clear, but it may arise from enhanced wall losses in the irradiated chamber unaccounted for in the experiments by Hallquist et al. (1997).





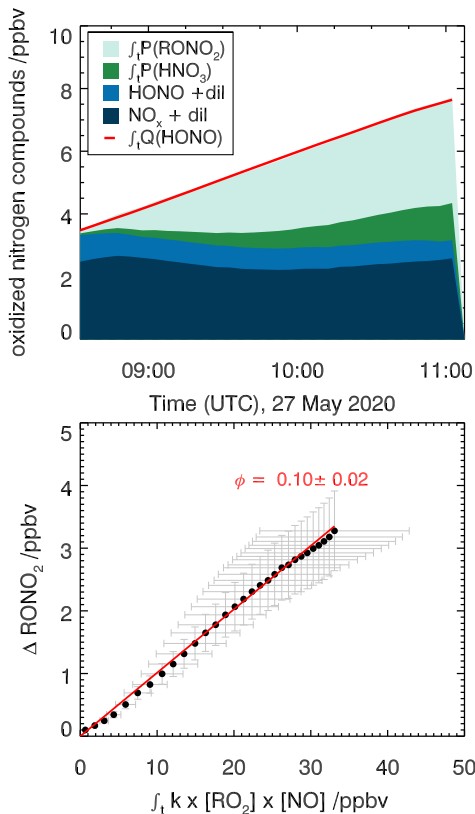

**Figure 10.** Determination of the organic nitrate yield for the reaction of caronaldehyde with OH for the experiment E6. HONO and $NO_x$ concentrations are measured, while $RONO_2$ and $HNO_3$ concentrations are calculated as described in the text.

### 3.3.3 Determination of alkyl nitrate yields for the reaction of caronaldehyde + OH

The nitrate yield from the reaction of caronaldehyde with OH is determined for experiment E6. The organic nitrate yield $\Phi_{RONO_2}$ is found to be $(10 \pm 2)\,\%$, following the procedure described in Sect. 2.5. As shown in Fig. 10, $3.5\,\mathrm{ppbv}$ of $RONO_2$ were formed in the chamber during the experiment E6 along with $1.2\,\mathrm{ppbv}$ of $HNO_3$. $NO_x$ and HONO concentrations were $2\,\mathrm{ppbv}$ and $1\,\mathrm{ppbv}$, respectively. The error is derived from the accuracies of measured concentrations. To our knowledge, this is the first report of a measured organic nitrate yield for the caronaldehyde + OH reaction.


Generally, the nitrate yield from aldehydes are typically smaller than those of monoterpenes. Organic nitrate yields for other aldehydes used in current chemical models like the MCM3.3.1 are $5\,\%$ for pinonaldehyde, the main daytime oxidation product of $\alpha$-pinene and $12\,\%$ for pentanal. Calculation of the organic nitrate yield from the caronaldehyde oxidation according to





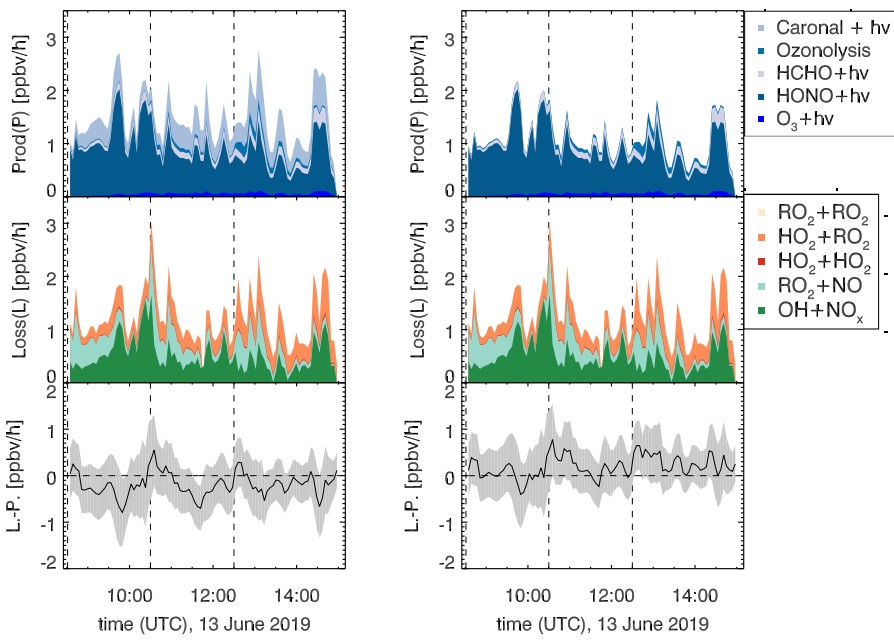

**Figure 11.** $RO_x$ production and destruction rates for the OH-induced photooxidation in the experiment E1, exluding (left) and including (right) the higher photolysis frequency determined in this work. Dashed lines indicate times, when trace gases were injected.

Jenkin et al. (2019) is $5.5\,\%$. The value determined in this study is within the range of nitrate yields reported in the literature,
but somewhat higher than values reported for similar aldehydes and calculations from SAR.

### 3.4   Production and destruction rates of $RO_x$ radicals in the oxidation of $\Delta^3$-carene

Field studies in forested environments, were $\Delta^3$-carene was one of the main monoterpenes emitted found that both measured
OH and $HO_2$ concentrations can not be reproduced by models (Kim et al., 2013; Hens et al., 2014). The analysis of radical
formation and loss recations for those environments revealed that a photolytic $HO_2$ source missing in the model calculations is
one possible explanation for the observed discrepancies. In the following, radical formation and loss rates of the sum of radicals
($RO_x$) and their differences are determined according to Equation (6) and (7). The results of this analysis for experiment E1
with NOx mixing ratios below $1\,\mathrm{ppbv}$ are shown in Fig. 11.

The turnover rates in this experiment reach maximum values between $2$ and $3\,\mathrm{ppbvh}^{-1}$. The main radical formation process
is HONO photolysis, contributing more than $70\,\%$ to the average formation rate. Assuming that the photolysis frequency is
higher by a factor of 7 (see Section 3.3.1), the formation of one $HO_2$ and one $RO_2$ radical per photolyzed caronaldehyde
molecule would have a higher influence on the radical budgets of the $\Delta^3$-carene + OH reaction than previuosly assumed.



This may therefore help to reduce the discrepancies between measured and modeled $HO_2$ concentrations in field studies. The photolysis of caronaldehyde contributes on average $0.3\,\mathrm{ppbvh^{-1}}$. $60\,\%$ of the average loss rate is due to the reaction of ROx
with NOx species. The remaining ROx losses can be explained by radical self-reactions, as can be expected for the experimental conditions because the reaction of the peroxy radical with $HO_2$ ($35\,\%$) or $RO_2$ radicals becomes competitive.

For experiments with NOx mixing ratios below $1\,\mathrm{ppbv}$ in the experiments in this work, the ROx budgets are closed within the uncertainty. It can therefore be assumed that there are no primary radical production or loss processes that are unaccounted for in this analysis. The contribution of the photolysis of caronaldehyde to the radical formation is in the range of the error of the
analysis, and the loss and formation of radicals would also be closed if this process was not considered. In the scope of this work, it is therefore not possible to distinguish whether the caronaldehyde photolysis will contribute significantly to reduce the previously observed discrepancies between measured and modeled $HO_2$ concentration in field studies. Possible reaction pathways leading to the formation and loss of radicals in the $\Delta^3$-carene oxidation are shown in Fig. 5. The reaction of $RO_2$ with $HO_2$ proceeds with a reaction rate constant of $1.9 \times 10^{-11}\,\mathrm{cm^3 molecule^{-1}s^{-1}}$ (Jenkin et al., 2019) mainly leading the
formation of hydroperoxide species ROOH. The reaction of $RO_2$ with $RO_2$ results in the formation of either a diol or an alkoxy radical RO, which again most likely decomposes forming caronaldehyde. The reaction rate constant of this reaction is calculated to be $1.2 \times 10^{-13}\,\mathrm{cm^3 molecule^{-1}s^{-1}}$ (Jenkin et al., 2019). In the experiments presented here, this reaction only contributes $< 1\,\%$ to the loss of total ROx radicals. Isomerization reactions like intramolecular H-shifts as investigated by Vereecken and Noziere (2020) can form a new pathway to product formation from $RO_2$. For the $RO_2$ described here, an
intramolecular 1,5-H-shift from the OH functional group to the peroxy-radical unit -OO would be the fastest of the possible isomerization reactions with a reaction rate constant of $3.5 \times 10^{-5}\,\mathrm{s^{-1}}$ expected from the SAR in (Vereecken and Noziere, 2020). Compared to the other described $RO_2$ loss rates, especially the loss in the reaction with NO ($0.07\,\mathrm{s^{-1}}$ at $0.3\,\mathrm{ppbv}$ NO) and $HO_2$ ($1.9 \times 10^{-3}\,\mathrm{s^{-1}}$ at $1.0 \times 10^8\,\mathrm{cm^{-3}}$ $HO_2$), this isomerization reaction is too slow to be signficant for atmospheric conditions like in the experiments here.

## 4 Summary and Conclusion

The photooxidation of $\Delta^3$-carene was investigated for $NO_x$ mixing ratios below $1.5\,\mathrm{ppbv}$ in the atmospheric simulation chamber SAPHIR. Photooxidation experiments were performed under atmospheric conditions with $\Delta^3$-carene mixing ratios in the range of $5$ to $7\,\mathrm{ppbv}$. In this study, the gas-phase organic nitrate yield of the $\Delta^3$-carene + OH reaction was determined for the first time and found to be $(25 \pm 4)\,\%$. The comparison of the obtained organic nitrate yield to yields obtained in the
aerosol phase (Rollins et al., 2010) and to the organic nitrate yield of the structurally similar monoterpene $\alpha$-pinene shows that the determined value is in accordance with the reported values. The reaction rate constant of the reaction of $\Delta^3$-carene with OH was determined to be $(8.0 \pm 0.5) \times 10^{-11}\,\mathrm{cm^3 s^{-1}}$ using an absolute rate technique approach. The obtained value was found to be in good agreement with reaction rate constants reported in the literature within the stated error. Additionally, the ozonolysis of $\Delta^3$ carene was studied. A reaction rate constant of $(4.4 \pm 0.2) \times 10^{-17}\,\mathrm{cm^3 s^{-1}}$ was found, with an OH yield
of $0.65 \pm 0.10$. The yield of the oxidation products caronaldehyde was determined for the $\Delta^3$-carene + OH and $\Delta^3$-carene +



O$_3$ reactions and found to be $(0.33 \pm 0.03)$ and $(0.055 \pm 0.02)$, respectively, in good agreement with reported literature values. The photolysis and OH-induced photooxidation of caronaldehyde were also studied. The photolysis frequency was calculated using the absorption spectrum measured by Hallquist et al. (1997), but it was found that to explain the observed caronaldehyde decay the absorption cross section would need to be higher by a factor of 7 assuming a maximum quantum yield of 1. The

caronaldehyde + OH reaction rate constant was found to be $(3.6 \pm 0.7) \times 10^{-11} \, \mathrm{cm^3 s^{-1}}$, in relatively good agreement with reported literature values. The experimental budget analysis of the loss and production processes of ROx radicals for the $\Delta^3$-carene + OH reaction shows that primary loss and production reactions are balanced within the uncertainty of the experiment of $\pm 0.5 \, \mathrm{ppbv h^{-1}}$. The formation of HO$_2$ and RO$_2$ radicals from the photolysis of caronaldehyde was considered as an additional radical source with the photolysis frequency determined in this work. The contribution of the photolysis reaction to the radical

formation was about 10 %. The fact that radical formation and loss reactions are well balanced indicates that there are no primary formation or loss processes unaccounted for in this analysis.

*Competing interests.* The authors declare to have no competing interests.

*Data availability.* Data of the experiemts in the SAPHIR chamber used in this work are available on the EUROCHAMP data homepage. 13 June 2019: https://doi.org/10.25326/BQNH-P286 21 May 2020: https://doi.org/10.25326/AGPD-MP70, 27 May 2020: https://doi.org/10.25326/8K59-

JC53, 30 May 2020: https://doi.org/10.25326/PXFM-3967, 31 May 2020: https://doi.org/10.25326/QACC-KJ83

*Author contributions.* LH and HF designed the experiments. AN and CC conducted the ROx radical measurements and the OH reactivity measurements. MG conducted the HONO measurements. RT, DR and SW were responsible for the PTR-TOF-MS and VOCUS measurements. BB conducted the radiation measurements. FR was responsible for the NO$_x$ and O$_3$ data. LH analyzed the data and wrote the paper with the help of HF. All co-authors commented and discussed the manuscript.

*Acknowledgements.* This project has received funding from the European Research Council (ERC) under the European Union's Horizon 2020 research and innovation programme (SARLEP grant agreement No. 681529) and from the European Commission (EC) under the European Union's Horizon 2020 research and innovation programme (Eurochamp 2020 grant agreement No. 730997).



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
