# Peer review of "Atmospheric photooxidation and ozonolysis of $\Delta^3$ -carene and 3-caronaldehyde: Rate constants and product yields"

_Atmospheric Chemistry and Physics, 2021_

## Author Comment (AC1)

*In this manuscript, the reaction rate constants of $\Delta^3$-carene with OH and O$_3$, OH yield in ozonolysis, and organic nitrate yield in photooxidation were first studied by combining model simulation and laboratory investigation. Caronaldehyde, as one of the main products of $\Delta^3$-carene oxidation, its yield from OH-initiated oxidation and photolysis, and the yield of organic nitrate were further calculated. The data obtained in the present study were compared with the corresponding values in the literature and nearly all the determined values are in good agreement with the reported values. Overall, the method is mostly reasonable and the uncertainty has been well considered. However, my main concern is what is the motivation of the focus on $\Delta^3$-carene? Just considering its atmospheric abundance? As demonstrated by the authors, the reaction mechanism of $\alpha$-pinene has been investigated in many studies and, $\Delta^3$-carene behaves similarly like $\alpha$-pinene regarding its oxidation mechanism. In the fact, the rate constants of the reaction of $\Delta^3$-carene with OH and O$_3$, the OH yield, as well as the organic nitrate yield have been already reported in the literature. A better introduction of the research background and motivation of learning the reaction rate constants and OH yield, as well as the significance of these studies are suggested. Overall though, this study could be published on Atmospheric Chemistry and Physics once the following comments are addressed.*

We would like to thank the reviewer for the feedback. The comments were very helpful to improve our manuscript.

**Comment 0:** *However, my main concern is what is the motivation of the focus on $\Delta^3$-carene? Just considering its atmospheric abundance? As demonstrated by the authors, the reaction mechanism of $\alpha$-pinene has been investigated in many studies and, $\Delta^3$-carene behaves similarly like $\alpha$-pinene regarding its oxidation mechanism.*
**Response:** Although there are several studies on some of the aspects of the reaction mechanism of $\Delta^3$-carene and caronaldehyde, the total number of studies is small and results of previous studies show partly large discrepancies. Our study investigated the chemical mechanism at atmospheric concentrations of reactants and at atmospheric conditions and with different methods than used before and thereby contributes to clarify which values are valid. In addition, the direct detection of radicals allowed to investigate, if so far unaccounted radical production is of importance in the atmospheric oxidation of $\Delta^3$-carene. Although carene is structurally similar to $\alpha$-pinene, one may not necessarily expect exactly the same behaviour as also demonstrated in the results. Therefore, we believe that our study can signficantly contribute to better understand the atmospheric oxidation of $\Delta^3$-carene, one important monoterpene emitted by plants. To address the raised concern also in the revised manuscript, the following line is added: "However, recent theoretical and experimental studies show that though the structural differences between the two molecules may be small, they may have an impact on the oxidation mechanism for example due to the impact of uni-molecular peroxy radical reactions (Draper et al., 2019; Moller et al., 2020)." (line 36-38)

**Comment 1:** *Page 4, line 85, what is the value of the detection limit of the instruments, e.g PTR-MS for the detection of $\Delta^3$-carene?*
**Response:** The LoD of the PTR instruments for monoterpenes is about $2\,\mathrm{ppb}$.

**Comment 2:** *Page 5, line 90, 30%, how the RH in Table 1 obtained? Is 30% the averaged RH measured at the beginning and at the end of the experiment? This is suggested to be noted in Table 1 by footnote.*
**Response:** Actually, there was a typo in the text. The realtive humidity at the start of the ozonolysis experiment E3 was 20% as denoted in Table 1. The typo is corrected in the revised version.

**Comment 3:** *Page 6, line 121, 'aproton', 'a proton'?*
**Response:** Has to be 'proton' without a, is corrected as suggested in the revised version.

**Comment 4:** *Page 7, line 165, line 165-167, why the fraction of $RO_2$ radicals produced by $\Delta^3$-carene to the total $RO_2$ concentration could be assumed to be equal to the ratio of OH-reactivity from the $\Delta^3$-carene + OH reaction to the total measured OH reactivity? As shown in equation 2, OH can react with $NO_2$ but without $RO_2$ generation. And if this assumption is right, why is $kOH_{VOC}$ but not $kOH_{obs}$ used in Equation 4?*
**Response:** $kOH_{VOC}$ is the fraction of the OH reactivity attributed to the reaction of OH with VOC species. It is obtained by

50 subtracting all contributions of inorganic species from the observed OH reactivity. Since all VOC species form an $RO_2$ radical upon the reaction with OH, the ratio of the calculated OH reactivity caused by Carene and the total OH recativity caused by VOCs species in general should be equivalent to the ratio of $RO_2$s formed from carene and from all other VOCs. Since $kOH_{obs}$ still includes contributions from inorganic reactions, $kOH_{VOC}$ is used as this is the sum of all OH reactions forming $RO_2$.

55 ***Comment 5:*** *Page 12, figure 2, which curve represents the time evolution of CO? The legend should be added.*
**Response:** The caption was misleading and is adapted according to the comments of Referee 1 and 2 in the revised version.

[Figure]

**. Figure 2.** Measured $\Delta^3$-carene, caronaldehyde (upper panel) and $O_3$ (lower panel) mixing ratios in the experiment investigating the ozonolysis of $\Delta^3$-carene (experiment E3). The shaded area indicates the time where CO was injected as an OH scavenger (experiment E4). Vertical lines give the times, when trace gases were injected.

**Comment 6:** *If the chamber was flushed with synthetic air until the concentrations of trace gases were below the detection limit of the instruments (Page 4), how this can be achieved for experiments E3 and E4? Figure 1 showed no time interval for such an operation.*

**Response:** The chamber was flushed over night prior the day-long experiments. Time series of trace gas concentrations in the figures only show time series from the point in time, when the target VOC was injected. Experiments E3 and E4 were done on the same day and flushing was only done once. To clarify this point, the following sentence is added to the revised manuscript: "Experiments E3 and E4 as well as Experiments E5 and E6 were conducted on the same day and flushing of the chamber was only done over night prior to experiment E3 and experiment E5, respectively." (line 87-89)

**Comment 7:** *Page 15, references for the ozonolysis mechanism is suggested to be cited*

**Response:** In this part, measured time series and experimental loss rates are used to determine the reaction rate constant and the OH yield. On this page, studies by Wang et al. (2019) and Aschmann et al. (2002) regarding the ozonolysis mechanism are cited in the revised version.

**Comment 8:** *Figure 4, What do the solid line and the dashed line represent? For* OH *oxidation, why there are different corrected caronaldehyde concentrations at one reacted $\Delta^3$-carene concentration?*

**Response:** Different experiments are considered for the caronalehyde yield from OH oxidation. The solid lines represent the slopes of the regression analysis, the dashed lines represent the error of the regression analysis. The caption of this figure is updated to clarify this in the revised manuscript.

[Figure]

**. Figure 4.** Yield of caronaldehyde for reactions R1 (OH+ $\Delta^3$-carene) and R2 (O$_3$+ $\Delta^3$-carene) determined from the slope of the relation between consumed $\Delta^3$-carene and measured caronaldehyde concentrations. For reaction R1 the result from experiments E1 and E2 are shown. Concentrations were corrected for dilution and chemical loss (see text for details). Caronaldehyde yields from the oxidation of $\Delta^3$-carene are determined as the slopes of the shown solid linear regression lines (black: OH oxidation (R1), green: ozonolysis (R2)). Dashed lines indicate the error of the regression analysis.

**Comment 9:** *Page 18, Figure 5, where is 35% derived from? The top 'HO$_2$' in this figure should be NO$_2$.*

**Response:** The 35% is calculated from the measured concentrations of these species in the chamber and their respective reaction rate constants with RO$_2$ as taken from MCM3.3.1. The NO$_2$ formed from the NO reaction of the RO$_2$ to form RO + NO$_2$ is added to the figure for clarification.

**Comment 10:** *Page 21, When 249, while the calculation of* RO$_{2,\text{carene}}$ *is reconsidered, the nitrate yield* $\phi_{\text{RONO}_2}$ *needs to be checked.*

**Response:** As outlined in our answer to comment 4 of referee 1, the calculation of RO$_{2,\text{carene}}$ does not have to be reconsidered. Therefore, we do not think that any changes in the manuscript are needed in response to this comment.

**Anonymous Referee #2**

90

*The manuscript describes results from chamber experiments to study the atmospheric fate of an important biogenic VOC, namely D-3-carene and one of its oxidation products, 3-caronaldehyde. The methods described appear to be robust, the results well supported by the data and the manuscript is commendably well-written and concise. My main concerns are around the presentation and descriptions (captions and legends) of some of the plots. I strongly recommend publication in ACP once these*

95 *fairly minor issues are resolved.*

We would like to thank the reviewer for careful reading and the feedback. The comments were very helpful to improve our manuscript.

100 *Comment 1: Fig 2 is missing legend information (what does the black line in the lower plot represent?) and potentially a dataset if we were really meant to see both $O_3$ and $CO$ time profiles.*
**Response:** True, the caption was misleading and is changed according to the comments in the revised version.

*Comment 2: On Fig 3, whilst a good match of experimental data to a yield of 0.65 is clear, the sensitivity of the system to*
105 *this parameter is less clear. Could additional dashed lines be used to indicate yields of, for example 0.25, 0.45, 0.85 to illustrate just how sensitive was the experiment?*
**Response:** An additional figure showing the sensitivity of the system to the applied yield is added in the Supplement. Additionally, a comment on the sensitivity is added in the main text: "The OH yield from ozonolysis is optimized until the measured decay of $\Delta^3$-carene matches the modeled decay (Fig. 3). A comparison of the experimental decay to model runs with different
110 OH yields is shown in Fig. S1." (line 321-322)

[Figure]

**. Figure S1** Measured and modeled $\Delta^3$-carene concentrations for the ozonolysis experiment E4. During the second part of the experiment (upper panel), 90 ppmv of an OH scavenger (CO) were injected to suppress OH, so that $\Delta^3$-carene reacted only with ozone. The modeled decay is fitted with a rate constant of $(4.4 \pm 0.2) \times 10^{-17}\,\mathrm{cm^3\,s^{-1}}$. During the first part of the experiment (lower panel), when no CO was present, the measured decay of $\Delta^3$-carene is signficantly faster than expected from ozonolysis alone. The measured time series of $\Delta^3$-carene can be best matched, if an OH yield of $0.65 \pm 0.10$ from its ozonolysis reaction is assumed. OH yields of 0.20, 0.40 and 0.80 are also presented in the lower panel to demonstrate the sensitivity of the system to changes in the OH yield.

*Comment 3: Fig 4 needs an improved caption. If I am interpreting it correctly, it need to state clearly that this plot was used to determine caronaldehyde yields from both (R1) and (R2).*

**Response:** The caption of Fig. 4 is changed according to the comment.

[Figure]

. **Figure 4.** Yield of caronaldehyde for reactions R1 (OH+ $\Delta^3$-carene) and R2 (O$_3$+ $\Delta^3$-carene) determined from the slope of the relation between consumed $\Delta^3$-carene and measured caronaldehyde concentrations. For reaction R1 the result from experiments E1 and E2 are shown. Concentrations were corrected for dilution and chemical loss (see text for details). Caronaldehyde yields from the oxidation of $\Delta^3$-carene are determined as the slopes of the shown solid linear regression lines (black: OH oxidation (R1), green: ozonolysis (R2)). Dashed lines indicate the error of the regression analysis.

115     *Comment 4: I am a little confused by what is meant by "RO2" in Fig 5. Does this refer to the peroxyl radical derived from carene + OH + O$_2$? If so, would the yields of both caronaldehyde and HO2 really be unity from the RO$_2$ + RO$_2$ channel? Could these yields be between 1 and 2? If "RO$_2$" is meant to represent other peroxyl radicals present in the chamber then the caption does not make sense when referring to additional pathways.*

**Response:** In the revised manuscript, Fig. 5 is adapted to clarify that point and the caption of Fig. 5 is changed to:

120

. **Figure 5** Simplified scheme of the first reaction steps of the OH photooxidation of $\Delta^3$-carene (adapted from Colville and Griffin (2004). Yields shown in black are from SAR by Jenkin et al. (2018). H-abstraction has only little influence on the presented product yields. $RO_2 + RO_2$ reactions could lead to other products than the formation of the alkoxy radical that produces caronaldehyde + $HO_2$, so that the yield of caronaldehyde from $RO_2 + RO_2$ reactions is expected to be less than 1.

***Comment 5:** One further point for clarification is regarding the rate coefficient determination for OH $+\Delta^3$-carene, of $k =$ (8.0 $\pm$ 0.5) cm3 s-1. I understand that a complex model was used to fit to experimentally determined decays of $\Delta^3$-carene and presumably to time profiles of other species. The resulting value is reported as the mean from several runs, each reporting an "optimised value". Do the uncertainties quoted (the $\pm$ 0.5) account for simply variability from one run to another (i.e. 0.5 is the standard error of the mean), or is there some extra uncertainty resulting from the optimisation process?*

**Response:** The given error refers to the variability from one run to the other. The optimization error is much smaller than the variablity of the results. A clarification regarding this point has been added to the manuscript: "The given error is the standard error of the mean. The optimization error was much smaller than the variability of the results in different experiments." (line 390-392)

***Minor comments:** Starting with abstract but also throughout - there should be space characters between units "cm3s-1" -> "cm3 s-1". Else it is easy to confuse units like m s-1 (metres per second) with ms-1 (permillisecond)*

**Response:** Changed according to the comment.

*Very minor point, but where % values are used there should be no space character, so "5 %" -> "5%" as % is part of the number itself, not a unit.*

**Response:** Changed according to the comment.

*There looks to me like a typo in the fourth Atkinson reference in the bibliography "O-x".*

**Response:** Changed according to the comment.

**References**

Aschmann, S. M., Arey, J., and Atkinson, R.: OH radical formation from the gas-phase reactions of $O_3$ with a series of terpenes, Atmos. Environ., 36, 4347–4355, https://doi.org/10.1016/s1352-2310(02)00355-2, 2002.

Colville, C. J. and Griffin, R. J.: The roles of individual oxidants in secondary organic aerosol formation from $\Delta$-3-carene: 1. gas-phase chemical mechanism, Atmos. Environ., 38, 4001–4012, https://doi.org/10.1016/j.atmosenv.2004.03.064, 2004.

Draper, D. C., Myllys, N., Hyttinen, N., Moller, K. H., Kjaergaard, H. G., Fry, J. L., Smith, J. N., and Kurten, T.: Formation of Highly Oxidized Molecules from $NO_3$ Radical Initiated Oxidation of $\Delta$-3-Carene: A Mechanistic Study, ACS Earth Space Chem., 3, 1460–1470, https://doi.org/10.1021/acsearthspacechem.9b00143, 2019.

Jenkin, M. E., Valorso, R., Aumont, B., Rickard, A. R., and Wallington, T. J.: Estimation of rate coefficients and branching ratios for gas-phase reactions of OH with aliphatic organic compounds for use in automated mechanism construction, Atmos. Chem. Phys., 18, 9297–9328, https://doi.org/10.5194/acp-18-9297-2018, 2018.

Moller, K. H., Otkjaer, R. V., Chen, J., and Kjaergaard, H. G.: Double Bonds Are Key to Fast Unimolecular Reactivity in First-Generation Monoterpene Hydroxy Peroxy Radicals, J. Phys. Chem. A, 124, 2885–2896, https://doi.org/10.1021/acs.jpca.0c01079, 2020.

Wang, L. Y., Liu, Y. H., and Wang, L. M.: Ozonolysis of 3-carene in the atmosphere. Formation mechanism of hydroxyl radical and secondary ozonides, Phys. Chem. Chem. Phys., 21, 8081–8091, https://doi.org/10.1039/c8cp07195k, 2019.

---

## Author Response (AR2)

**Anonymous Referee #1: Suggestions for revision**

*Comment 1: 1. Page 6, line 87, though the LoD of PTR has been stated by the authors, but the LoD for other instruments, including those used for the measurement of HCHO, HONO, NOx, O3. These values are expected as experiments were con-*
5  *ducted with atmospheric relevant concentrations of reactants and at atmospheric conditions.*
**Response:**The LoD of all used instruments was added to Table S1 in the Supplement.

**Table 1.** Instrumentation for radical and trace gas detection in the experiments.

| measured quantity | measurement technique | time resolution | accuracy $(1\,\sigma)$ | LoD $(1\,\sigma)$ |
|---|---|---|---|---|
| OH | laser-induced fluoresence (LIF) | $47\,\mathrm{s}$ | $13\,\%$ | $0.7\times10^6\,\mathrm{cm}^{-3}$ |
| $HO_2$, $RO_2$ | laser-induced fluoresence (LIF) | $47\,\mathrm{s}$ | $16\,\%$ | $0.8\times10^7\,\mathrm{cm}^{-3}$ |
| $k_{\mathrm{OH}}$ | laser photolysis + LIF | $180\,\mathrm{s}$ | $10\,\%$ | $0.3\,\mathrm{s}^{-1}$ |
| $\Delta^3$-carene | proton-transfer-reaction mass-spectrometer | $40\,\mathrm{s}$ | $7\,\%$ | $2\,\mathrm{pptv}$ |
| CO | cavity ring-down spectroscopy | $60\,\mathrm{s}$ | $1\,\mathrm{ppbv}$ | $80\,\mathrm{ppbv}$ |
| NO | chemiluminescence | $180\,\mathrm{s}$ | $5\,\%$ | $4\,\mathrm{pptv}$ |
| $NO_2$ | chemiluminescence | $180\,\mathrm{s}$ | $5\,\%$ | $2\,\mathrm{pptv}$ |
| HONO | long-path absorption photometry | $300\,\mathrm{s}$ | $20\,\%$ | $5\,\mathrm{pptv}$ |
| $O_3$ | UV-absorption | $10\,\mathrm{s}$ | $5\,\%$ | $1\,\mathrm{ppbv}$ |
| HCHO | Hantzsch monitor | $90\,\mathrm{s}$ | $8.5\,\%$ | $0.1\,\mathrm{ppbv}$ |
| HCHO | cavity ring-down spectroscopy | $300\,\mathrm{s}$ | $1.5\,\mathrm{ppbv}$ | $0.1\,\mathrm{ppbv}$ |
| photolysis freq. | spectroradiometer | $60\,\mathrm{s}$ | $10\,\%$ | [a] |

[a] several orders of magnitude lower than the maximum value at noon

*Comment 2:2. The atmospheric relative humidity is much higher than 20%. Why RH for the photooxidation experiments were around 80% at the beginning of the experiment while it was 20% for the ozonolysis experiments?*
10  **Response:**During the photochemistry experiments water vapour is needed for the production of OH radicals from the photolysis of ozone. The yield of OH radicals only depends on the absolute water mixing ratio and not on relative humidity. No other water vapour effect is expected. Over the course of the experiment in the illuminated chamber the relative humidity decreased to much lower values than the initial value of 80% due to the increase in temperature. During the ozonolysis experiments, water vapour is not expected to play a role for the chemistry investigated in this work. Therefore, the difference in the initial relative
15  humidity in the photochemistry experiments and the ozonolysis experiments does not play a role for the results of this work.